# Interplay of disordered and ordered regions of a human small heat shock protein yields an ensemble of 'quasi-ordered' states

Amanda F Clouser[1], Hannah ER Baughman[1,2], Benjamin Basanta[1], Miklos Guttman[2], Abhinav Nath[2], Rachel E Klevit[1]*

[1]Department of Biochemistry, University of Washington, Seattle, United States; [2]Department of Medicinal Chemistry, University of Washington, Seattle, United States

**Abstract** Small heat shock proteins (sHSPs) are nature's 'first responders' to cellular stress, interacting with affected proteins to prevent their aggregation. Little is known about sHSP structure beyond its structured α-crystallin domain (ACD), which is flanked by disordered regions. In the human sHSP HSPB1, the disordered N-terminal region (NTR) represents nearly 50% of the sequence. Here, we present a hybrid approach involving NMR, hydrogen-deuterium exchange mass spectrometry, and modeling to provide the first residue-level characterization of the NTR. The results support a model in which multiple grooves on the ACD interact with specific NTR regions, creating an ensemble of 'quasi-ordered' NTR states that can give rise to the known heterogeneity and plasticity of HSPB1. Phosphorylation-dependent interactions inform a mechanism by which HSPB1 is activated under stress conditions. Additionally, we examine the effects of disease-associated NTR mutations on HSPB1 structure and dynamics, leveraging our emerging structural insights.

DOI: https://doi.org/10.7554/eLife.50259.001

*For correspondence:
klevit@uw.edu

Competing interests: The authors declare that no competing interests exist.

## Introduction

Small heat shock proteins (sHSPs) are a class of molecular chaperones that help maintain cellular proteostasis. Like other heat shock proteins, sHSPs are believed to interact with exposed hydrophobic regions of partly unfolded or misfolded proteins to help prevent irreversible aggregation, but unlike other heat shock proteins, they perform their functions independent of ATP. sHSPs are implicated in numerous human diseases on the basis of inherited mutations in the protein sequence or upregulation in certain cancers (*Datskevich et al., 2012*). Cellular stressors such as oxidation and acidosis can influence their function (*Clouser and Klevit, 2017*; *Rajagopal et al., 2015a*; *Chernik et al., 2004*; *Alderson et al., 2019*), and stress-induced phosphorylation of sHSPs typically increases their chaperone activity (*Rogalla et al., 1999*; *Koteiche and McHaourab, 2003*; *Ahmad et al., 2008*). Despite their important roles in health and disease, relatively little is known about sHSP structure or structure-function relationships compared to other classes of chaperones.

Several properties make mammalian sHSPs particularly challenging for structural characterization. First and foremost, many sHSPs form large homo- and/or hetero-oligomers, and many of these exist as broad distributions of oligomeric species that contain different numbers of subunits. Furthermore, subunits exchange rapidly among oligomers, making it difficult to capture a single oligomeric state, or even a narrow distribution of states. Second, up to 50% of the sequence of sHSPs is believed to be intrinsically disordered and is unresolved in the few available structures of full-length sHSPs.

Third, there is often local conformational heterogeneity even within ordered regions of the protein. Finally, the transient and promiscuous nature of interactions between sHSPs and client proteins makes it challenging to capture a sHSP/client complex of the sort that has fueled structural-mechanistic understanding of other chaperones. Our goal has been to develop hybrid strategies capable of providing structural/dynamical residue-level information on these challenging yet highly important systems.

sHSPs consist of three regions that also define different types of inter-subunit interactions in homo-oligomers (*Figure 1A*). Most current structural information on human sHSPs is based on the structures of their defining feature, the central α-crystallin domain (ACD) (*Rajagopal et al., 2015a*; *Bagnéris et al., 2009*; *Clark et al., 2011*; *Baranova et al., 2011*; *Rajagopal et al., 2015b*; *Hochberg et al., 2014*). On their own, ACDs form dimers that are amenable to solution NMR and protein crystallography. Both sequentially and structurally conserved among the human paralogs, ACDs form an IgG-like β-sandwich fold of six or seven strands in truncated (ACD-only) structures and in the few oligomeric models and structures determined to date (*Figure 1B*) (*Jehle et al., 2011*; *Braun et al., 2011*; *Clark et al., 2018*). ACDs dimerize by anti-parallel alignment of their β6+7 strands to form a long β-sheet dimer interface (ACD-ACD interaction). The flanking regions on either side of the ACD are far less conserved among paralogs and are predominantly disordered. The C-terminal region (CTR) is a relatively short extension that contains many charged residues and is highly flexible and disordered. CTRs are thought to serve as solubility tags for sHSPs. Most human sHSPs known to exist as large oligomers contain a three-residue motif of alternating Ile or Val residues known as the 'IXI' motif near the beginning of their CTR. IXI motifs can bind in a 'knob-and-hole' fashion into a groove formed between the top and bottom sheets of the ACD β-sandwich (the 'β4/β8 groove', *Figure 1B*), defining a second type of intermolecular interaction observed in sHSP oligomers (ACD-CTR interaction). Finally, the N-terminal region (NTR) is a relatively long extension (50–100 residues in vertebrates) that is presumed to be disordered based on secondary structure prediction and the lack of density in electron microscopy (EM) and X-ray crystallography-based structures of sHSPs (*McHaourab et al., 2012*; *Kim et al., 1998*; *White et al., 2006*). Relative to typical disordered regions, the NTR is enriched in hydrophobic and aromatic residues. Short regions of order have been observed in the NTR of HSPB5 by solid-state nuclear magnetic resonance (NMR) (*Jehle et al., 2011*) (*Figure 1—figure supplement 1*), in a crystal structure of HSPB6 in complex with a client protein (*Sluchanko et al., 2017*), and in a crystal structure of an HSPB2/HSPB3 heterotetramer (*Clark et al., 2018*). The lack of sequence conservation in the NTRs among paralogs (*Figure 1C*) raises the question of whether there are similarities among structural features of NTRs of other sHSPs or whether each protein is idiosyncratic. As the NTR drives sHSP oligomerization and is thought to play a critical role in its chaperone activity, the paucity of structural insights has greatly slowed progress towards understanding these important proteostasis components.

Among the most ubiquitously expressed of the ten human sHSPs, HSPB1 is implicated in multiple biological roles in many cellular pathways, diseases, and tissues. In addition to its canonical chaperone role, HSPB1 has been implicated in the apoptosis pathway and interacts with cytoskeleton proteins (*Gusev et al., 2002*; *Lanneau et al., 2008*). Upregulation of HSPB1 has been observed in certain types of cancer and has therefore drawn attention as a potential therapeutic target (*Gibert et al., 2013*). HSPB1 is phosphorylated within minutes after cells are subjected to stress by stress-related kinases at three NTR serine residues that are conserved among HSPB1 orthologs (*Figure 1A,C*, and *Figure 1—figure supplement 1*) (*Larsen et al., 1997*). Biochemical investigations have demonstrated that phosphorylated HSPB1 or phosphomimetic mutants form oligomers that are much smaller and more active than the distribution formed by unmodified HSPB1 (*Lambert et al., 1999*; *Hayes et al., 2009*; *Lelj-Garolla and Mauk, 2005*; *McDonald et al., 2012*; *Jovcevski et al., 2015*). Several inherited mutations in HSPB1 are reported to be associated with the severe neurological disorders Charcot-Marie-Tooth disease and distal hereditary motor neuropathy. Disease-associated mutations are harbored in all three regions of the protein, with overrepresentation in the NTR and in residues near the dimer interface of the ACD, and most appear to be autosomal dominant. Little is known about the structural mechanism for phosphorylation-dependent dispersion of HSPB1 into smaller oligomers or the structural effects of disease-associated mutations.

Here, we present the first residue-level structural analysis of full-length HSPB1, using a hybrid experimental approach of NMR spectroscopy and hydrogen-deuterium exchange mass spectrometry (HDXMS). Our study sheds light on the structural consequences of stress-induced phosphorylation

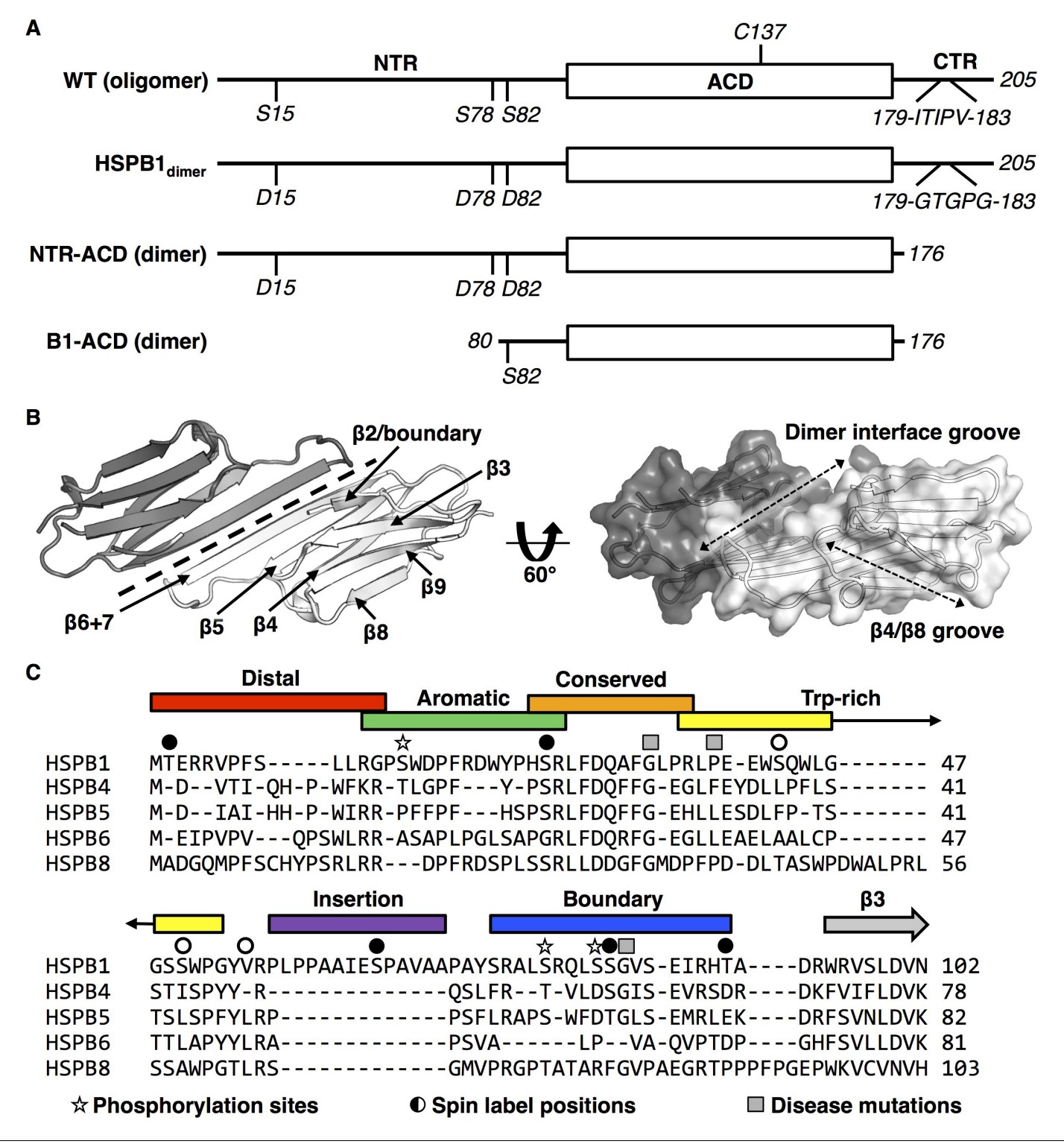

**Figure 1.** sHSP domain architecture and sequence alignment. (**A**) All sHSPs are defined by a conserved α-crystallin domain (ACD), that is flanked by N-terminal and C-terminal regions (NTR and CTR, respectively). Constructs used in this study are shown and their quaternary structure (oligomer vs. dimer) is indicated. HSPB1 contains a single cysteine, located in the ACD, which was substituted with serine in NMR-PRE and HDXMS experiments across several constructs. Phosphorylation-mimicking mutations in the NTR along with mutation of the IXI motif in the CTR (to abrogate ACD-CTR interactions) yields a full-length dimer, used in HDXMS experiments here. The C-terminally truncated construct (NTR-ACD) was used predominantly for NMR assignments and PRE experiments. The N-terminally and C-terminally truncated B1-ACD construct was used for peptide-binding NMR

*Figure 1 continued on next page*

*Figure 1 continued*

experiments. (**B**) ACD homodimer structure (4MJH), with dotted line indicating axis of symmetry along which ACDs interact via β6+7 strands. Dotted arrows in the right panel indicate the axes of the dimer interface and β4/β8 grooves. (**C**) NTR alignment of better-characterized human sHSPs shows minimal sequence conservation aside from the 'conserved' region. The NTR sequence of HSPB1 was divided into six sub-regions for this study, which were probed in peptide form (with the exception of the highly hydrophobic insertion region). Sites of interest in this study are also indicated: 1) phosphorylation sites mutated to aspartate to mimic phosphorylation, 2) sites that were targeted for spin label attachment, and 3) three disease-associated mutations (G34R, P39L, and G84R).

DOI: https://doi.org/10.7554/eLife.50259.002

The following figure supplement is available for figure 1:

**Figure supplement 1.** Sequence alignment of HSPB1 orthologs, residues of interest, and known and predicted secondary structure.

DOI: https://doi.org/10.7554/eLife.50259.003

and disease-associated mutations in HSPB1. We focus first on a dimeric species that represents a fully dissociated, stress-activated form of HSPB ('HSPB1$_{dimer}$') that is as effective at delaying the aggregation of the client protein, tau, as oligomeric wild-type (WT) HSPB1 (*Baughman et al., 2018*). The dimer is generated by three phosphorylation-mimicking substitutions in the NTR (S15D, S78D, and S82D) and substitution of the CTR IXI motif to 'GXG' ($^{179}$ITIPV$^{183}$ to GTGPG) (*Figure 1A*). Due to its monodisperse nature and relatively low molecular weight, the construct is amenable to residue-level analysis by solution-state NMR. Application of HDXMS to HSPB1$_{dimer}$ revealed regions of the NTR that are protected from exchange, consistent with the presence of transient order. We defined sub-regions of the NTR based in part on HDXMS behavior, as illustrated in *Figure 1C*, and performed two types of NMR experiments. NMR binding experiments using peptides containing the NTR sub-regions and PRE experiments in which spin labels were placed in each sub-region. Together, these approaches revealed that, although the NTR is predominantly disordered, there are nevertheless specific interactions between the NTR and ACD in the context of full-length HSPB1. HDXMS data obtained for WT-HSPB1 oligomers show remarkably similar patterns of protection, consistent with the interactions and order defined in the HSPB1$_{dimer}$ being conserved in the context of oligomeric HSPB1. Finally, HDXMS analysis of disease-associated mutations in HSPB1 reveals that the NTR is highly sensitive to single residue changes, resulting in non-local structural effects.

Altogether, our results show that even in a monodisperse form of HSPB1, there is substantial conformational heterogeneity, with multiple, specific contacts between regions of the NTR and the ACD. These contacts are altered in activation-mimicking and disease-associated mutated states, shedding light on the mechanisms by which perturbations such as phosphorylation or mutation can influence sHSP structure and function. The experimental approach presented here can be applied to other structurally heterogeneous systems that have proven difficult to study by traditional means, particularly those containing a mixture of ordered and disordered regions.

## Results

### The disordered NTR makes extensive contacts with the ACD

Atomic-level structural information for HSPB1 ACD and CTR regions is available from crystallographic and NMR studies (*Baranova et al., 2011*; *Rajagopal et al., 2015b*; *Hochberg et al., 2014*; *Alderson et al., 2017*). Given the crucial yet enigmatic role of the disordered NTR in sHSP oligomerization and function, we sought to expand structural studies to include the NTR. Although oligomeric forms of HSPB1 are too large to analyze by traditional solution-state NMR and are too heterogeneous to crystallize, our previously reported HSPB1$_{dimer}$ is amenable to solution-state NMR approaches (*Baughman et al., 2018*). A construct in which the CTR is truncated to the same position as the end of our B1-ACD construct (residue 176) is also dimeric in the phosphomimic context (termed 'NTR-ACD', *Figure 1A*). Thus, HSPB1$_{dimer}$ and its truncated form provide the first opportunity to obtain residue-level information of a sHSP with its NTR in solution. The simplest model for HSPB1$_{dimer}$ would be a structured ACD dimer with flexible, disordered NTRs and CTRs that behave independently of other domains. Such a species would give rise to an NMR spectrum that would resemble that of the isolated ACD plus resonances that correspond to 'random coil' positions. The NMR spectrum of an HSPB1 construct that lacks both its NTR and CTR ('B1-ACD', *Figure 1A*) and

forms a well-structured dimer has been assigned (*Rajagopal et al., 2015b*). Remarkably, few peaks overlap perfectly in overlays of $^1H$-$^{15}N$ HSQC spectra of B1-ACD and NTR-ACD (black versus blue, respectively; *Figure 2A*), Therefore, the model in which the ACD behaves independently of its flanking domains is inaccurate, demanding investigation of interactions between domains.

The NMR spectrum of NTR-ACD overlays well with that of full-length HSPB1$_{dimer}$, confirming that the CTR lacking its IXI motif does not interact detectably with the rest of the protein in the otherwise phosphomimetic context (*Figure 2—figure supplement 1*). NTR-ACD was therefore used for subsequent NMR studies to limit spectral overlap of CTR peaks with NTR peaks in the central portion of the HSQC spectrum corresponding to disordered residues. Most peaks corresponding to ACD residues could be identified and assigned from standard heteronuclear triple resonance NMR spectra collected on NTR-ACD (*Figure 2—figure supplement 1*, assignments deposited in BMRB). Despite the lack of precise overlap between the spectra of NTR-ACD and B1-ACD, the majority of residues in both contexts give rise to similar chemical shifts ($^1H$, $^{15}N$, and $^{13}C$). The similarities of the chemical shift 'fingerprints' indicate that the ACD structure is retained in the two contexts. Therefore, the widespread $^{15}NH$ chemical shift perturbations (CSPs) observed for ACD resonances indicate differences in environment due to proximity of the NTR.

The largest perturbations in ACD peaks are for residues in the β3, β4, and β8 strands and loop L3/4 (*Figure 2B*). These structural elements compose two grooves on the ACD dimer, known as the β4/β8 groove and the β3/β3 or 'dimer interface' groove (*Figure 1B*). In some cases, resonances appear to be absent in the NTR-ACD spectrum altogether: we were unable to identify peaks for several residues despite having assignments for these in the B1-ACD spectrum (gray squares in *Figure 2B*). Analysis of 3D-heteronuclear spectra used to assign the NTR-ACD spectrum failed to identify peaks with similar $^{13}C_\alpha$/$^{13}C_\beta$ chemical shifts for these residues, implying that resonances for these residues are not observed and are likely undergoing intermediate exchange between different conformations and/or chemical environments. As there is no evidence of slow chemical exchange in the context of B1-ACD, the broadening is likely due to dynamics and/or heterogeneity arising from the NTR. Altogether, the large perturbations indicate that the NTR interacts with the β4/β8 grooves at either end of the ACD dimer and the dimer interface at the center of the ACD.

Some ACD residues have multiple peaks in the NTR-ACD spectrum, indicating that they populate different chemical environments that interconvert slowly. The smallest difference in frequencies observed between multiple peaks from a single residue is 2 Hz, indicating a lifetime greater than 500 milliseconds. Notably, no peak doubling is observed in the B1-ACD spectrum. We could assign multiple peaks for four residues: positions 110 and 114 (flanking the β4 strand), position 148 (in loop L7/8, preceding the β8 strand), and residue 123 (in loop L5/6 between β5 and β6+7 strands). These positions are all near regions that exhibit CSPs and/or exchange broadening described above.

To investigate sources of the perturbations in ACD peaks in the NTR-ACD context, we examined NMR spectra of a mixture of $^{15}N$-B1-ACD and unlabeled NTR-ACD. Under reducing conditions (to inhibit disulfide bond formation at the dimer interface), mixed dimers composed of one $^{15}N$-B1-ACD subunit and one NTR-ACD subunit can form. This allowed us to observe perturbations in an ACD due to the NTR of its dimeric binding partner. As shown in *Figure 2A*, new peaks are observed in the spectrum of the mixture (red spectrum) that align with peaks in the NTR-ACD spectrum (blue spectrum). Peaks that exhibit this behavior nearly all correspond to residues in the β4/β8 groove and its flanking loops, predominantly the same residues whose peaks exhibit the largest CSPs compared to their position in the ACD-only spectrum (*Figure 2B*). The congruence of certain peaks in the mixed-dimer spectrum with those in the NTR-ACD spectrum provides unambiguous evidence that their resonance positions arise from an interaction involving the NTR of the other subunit within a dimer. The identity of these peaks reveals that a site of interaction is the β4/β8 groove and its flanking loops. Additionally, many other peaks in the mixed-dimer spectrum lose intensity, shift, and/or change shape as compared to the B1-ACD spectrum. Such behavior is indicative of increased heterogeneity caused by multiple processes. Peaks that undergo such changes correspond to residues in the β3 strand and preceding residues and at the end of the β9 strand and start of the CTR (*Figure 2—figure supplement 2*). Overall, the extent of perturbations observed in this mixing experiment reveals wide scale NTR-ACD contact and establishes that NTR/ACD interactions can occur between subunits within a dimer.

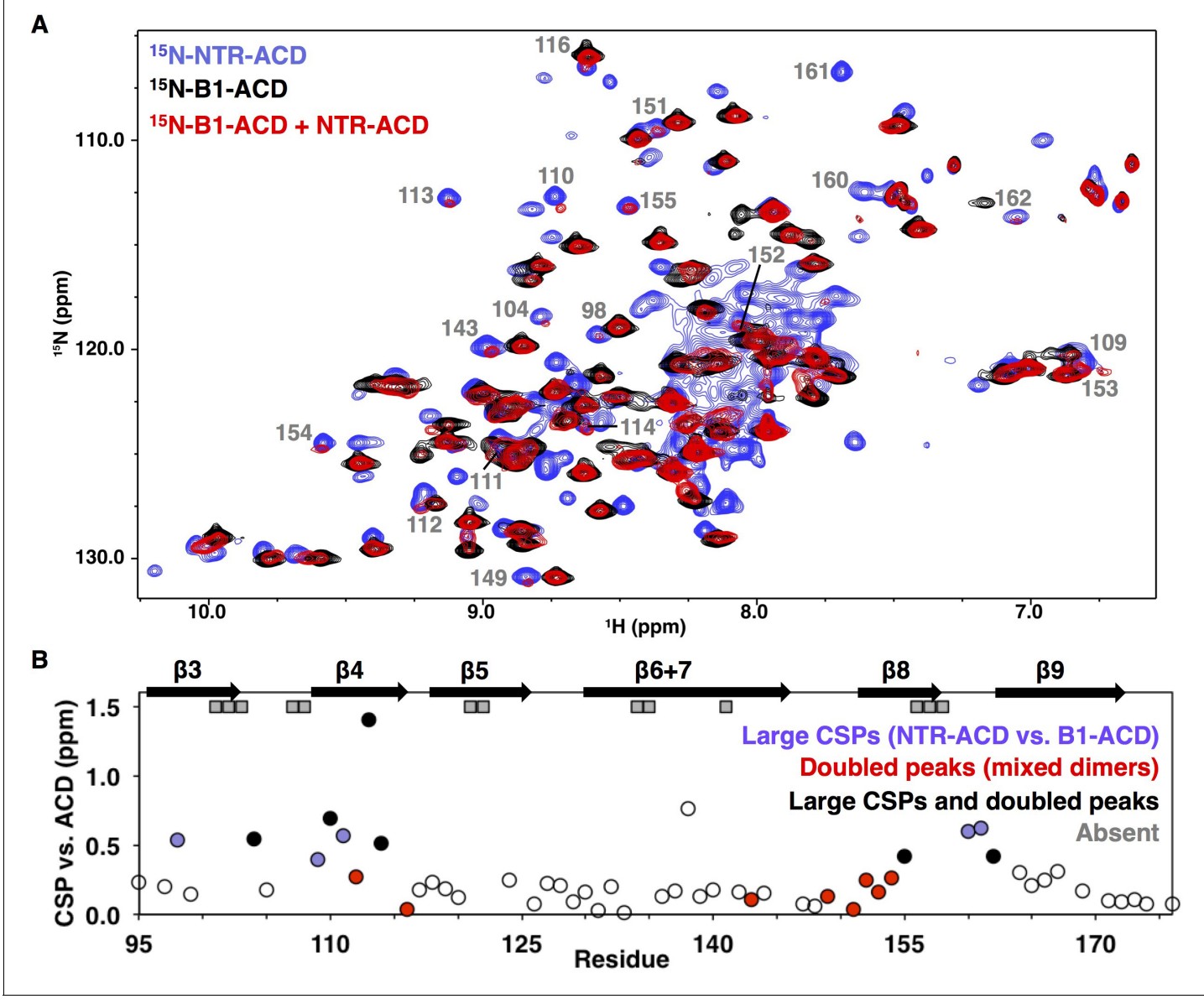

**Figure 2.** NMR analysis of NTR-ACD reveals changes in the ACD and increased heterogeneity when the NTR is present. (A) Comparison of B1-ACD (black), NTR-ACD (blue), and a mixture of $^{15}$N-B1-ACD and unlabeled NTR-ACD (red) $^{1}$H-$^{15}$N HSQC-TROSY spectra. (B) CSPs measured for NTR-ACD compared to B1-ACD. The following color-coding highlights regions of interest: blue, residues most perturbed in NTR-ACD compared to B1-ACD; red, residues that show NTR-ACD-like chemical shifts in the ACD mixing experiment; black, residues that show effects for both cases. These peaks are indicated in the spectrum. Gray squares correspond to residues in NTR-ACD whose resonances are missing and are presumably in substantially different chemical environments between B1-ACD and NTR-ACD.

DOI: https://doi.org/10.7554/eLife.50259.004

The following figure supplements are available for figure 2:

**Figure supplement 1.** NMR spectra comparison of HSPB1$_{dimer}$ and NTR-ACD.

DOI: https://doi.org/10.7554/eLife.50259.005

**Figure supplement 2.** Perturbed residues in the mixed $^{15}$N-B1-ACD/NTR-ACD dimer spectrum.

DOI: https://doi.org/10.7554/eLife.50259.006

## Distinct regions of the NTR bind different ACD interfaces

To ask whether specific NTR regions interact with the ACD, we subdivided the NTR sequence into six sub-regions, referred to here as the distal region, aromatic region, conserved motif, tryptophan-

rich region, insertion, and boundary region (*Figure 1C*). We used NMR to test the ability of peptides from each region to bind to B1-ACD. As described below, peptides from the distal, aromatic, conserved motif, and boundary regions all caused distinct changes (either chemical shift perturbations, loss of peak intensity, or both) in the $^{15}$N-HSQC spectrum of B1-ACD, implying specific interactions of these regions with the ACD. No perturbations were observed for the Trp-rich peptide, providing confidence that the effects observed for other peptides are specific.

We also performed paramagnetic relaxation enhancement (PRE) experiments to detect interactions between the NTR and the ACD within the context of the NTR-ACD construct. PRE from a spin label broadens resonances of residues proximal to the label. The sole native cysteine at position 137 was substituted with serine and multiple constructs were created in which a single cysteine residue was introduced at NTR sites to which the spin label MTSL was conjugated. Spin label positions were selected to probe each NTR region, as shown in *Figure 1C*. Before performing NMR experiments, each Cys mutant was assessed for alterations in secondary structure and oligomeric properties (by CD and SEC, respectively, data not shown). Only variants that retain HSPB1$_{dimer}$-like properties were investigated further. $^{15}$N and spin label were incorporated into these species and HSQC spectra of labeled dimers were collected with the active spin label and with the MTSL quenched by ascorbate. Quenched spectra were compared to the NTR-ACD spectrum to confirm that MTSL conjugation did not significantly perturb the constructs. PREs were quantified as ratios of peak intensity in active vs. quenched MTSL spectra. A range of behaviors was observed, depending on the spin label position: 1) strong, discrete effects, 2) smaller but still localized effects, or 3) widespread general effects.

As summarized in *Figure 3*, the results from the PRE experiments agree well with the peptide-binding studies, confirming that the interactions observed in the peptide-binding studies are recapitulated in the full-length protein. Results from individual peptides and spin-labels are described below and are provided in *Figures 4*, *5* and *Figure 4—figure supplement 1*.

Addition of the distal peptide to $^{15}$N-B1-ACD caused distinct chemical shift perturbations (CSPs) in the $^{15}$N-HSQC spectrum that map to the β4/β8 groove (*Figures 3*, *4A and B*). Importantly, peaks shift along a trajectory between their positions in the spectra of B1-ACD and NTR-ACD (as indicated by the arrows in *Figure 4A*). This is strong evidence that the large perturbations observed for these peaks in the NTR-ACD spectrum (relative to B1-ACD) are due to binding of the NTR distal region to the β4/β8 groove. The peptide binding was not saturated in the NMR experiments, thereby producing smaller CSPs relative to the NTR-ACD spectrum. The effective concentration of the peptide when attached to the ACD in its native form will be in the millimolar range, so the effects observed when the peptide is added in *trans* are relevant to the native state. The results add clarity to the $^{15}$N-B1-ACD/NTR-ACD mixing experiment described above (*Figure 2A*). The new peaks that appear in positions that correspond to those observed in the NTR-ACD spectrum are due to distal region binding to the β4/β8 groove of the other subunit of the dimer in a 'domain swap' relationship. We cannot rule out the possibility of a similar intra-chain interaction, but if it occurs it must be essentially identical to the inter-chain one observed in the mixed dimer.

The spectrum of NTR-ACD with MTSL at position two has strong peak intensity loss in two distinct sequence regions that correspond to loops L7/8 and L4/5, both of which lie near one entrance to the β4/β8 groove (*Figures 3* and *5A*). Other peaks in the spectrum are largely unaffected by the spin label (i.e., $I_{para}/I_{dia} \sim 1.0$). This remarkably discrete PRE effect from a spin label at the extreme N-terminus of HSPB1 indicates that when the region is near the ACD, it inhabits a highly localized position. Furthermore, the PREs are consistent with only one of the two possible orientations of distal region binding in the groove, namely aligned parallel to the β8 strand and antiparallel to the β4 strand such that position two only contacts residues near the beginning of β8 and the end of β4. This is the opposite orientation from that observed for the CTR IXI motif bound in the β4/β8 groove observed in a crystal structure of HSPB1-ACD (4MJH) (*Hochberg et al., 2014*). Distal region binding to the β4/β8 groove has been observed in crystals of HSPB6 (*Sluchanko et al., 2017*) and an HSPB2/3 (*Clark et al., 2018*) complex, but those sHSPs contain canonical IXI motifs in their NTRs. HSPB1 does not contain such a motif in its distal region; we propose instead that alternating hydrophobic residues in the HSPB1 segment $^6$VPFSLL$^{11}$ bind, supporting the idea that other hydrophobic amino acids can participate in β4/β8 binding. Our results thus identify a novel interaction and indicate that motifs from both the NTR and CTR of HSPB1 can bind in the β4/β8 groove but are oriented in opposite directions within the groove.

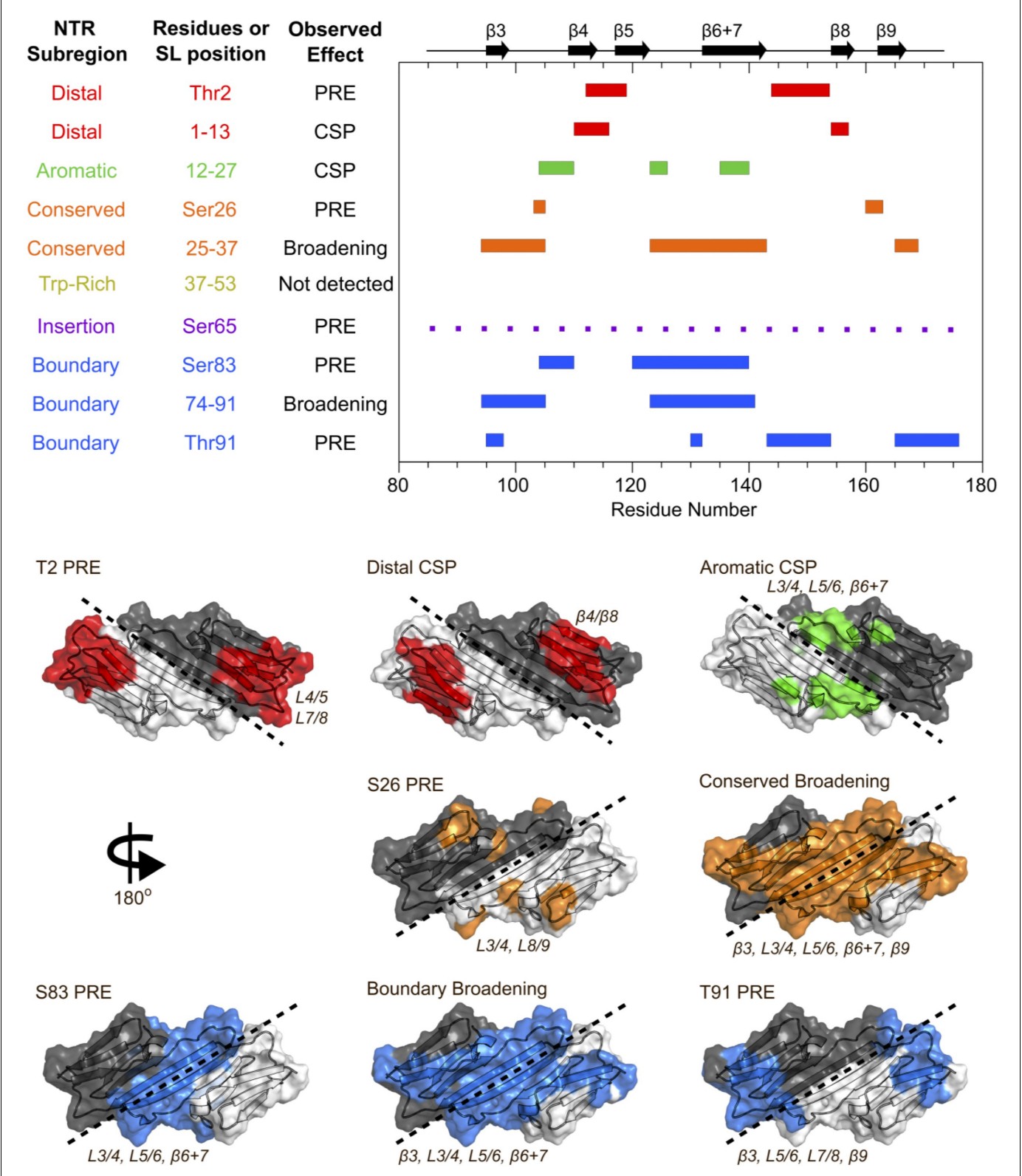

**Figure 3.** Summary of results from peptide-binding and PRE NMR experiments. Regions of the ACD that are perturbed upon peptide binding or lose intensity in the presence of a spin label are highlighted on the HSPB1 primary structure (top) and NMR structure (PDB 2N3J, bottom). The peptide from the Trp-rich region did not cause significant perturbations, and the presence of a spin label at position 65 caused widespread, nonspecific intensity loss, so these were not included in the bottom panel.

*Figure 3 continued on next page*

*Figure 3 continued*

DOI: https://doi.org/10.7554/eLife.50259.007

Peptide binding and PRE results indicate that the conserved motif binds at the ACD dimer interface groove (i.e. strands β3 and β6+7 from each subunit), and that the aromatic sub-region spans between the β4/β8 groove and the dimer interface, connecting the distal region to the conserved motif. The aromatic peptide (residues 12–27) caused distinct chemical shift perturbations in peaks corresponding to one side of the ACD, primarily loops L3/4 and L5/6, as well as some residues in β6 +7 (*Figures 3*, *4C and D*). This is consistent with the orientation inferred for the distal region, and shows that as the NTR exits the β4/β8 groove, it contacts the side of the ACD via loops L3/4 and L5/6 (*Figure 3*). The aromatic region contains one of the three HSPB1 phosphorylation sites (Ser15, *Figure 1C*), so we also tested binding by a version of the aromatic peptide containing phospho-serine at position 15. For the most part, the CSPs are absent or markedly reduced (*Figure 4—figure supplement 1A*), indicating that phosphorylation of Ser15 serves to release the aromatic region from the ACD. We note that both L3/4 and L5/6 are enriched in negatively-charged amino acids ($^{100}$DVNHFAPDE and $^{124}$HEERQDEHG, respectively), suggesting an electrostatic mechanism by which phosphorylation at Ser15 could regulate HSPB1 structure and activity.

Conserved motif binding at the ACD dimer interface is indicated by intensity loss in peaks of residues in β3, β6+7, and in loop L3/4 upon addition of the conserved peptide to $^{15}$N-B1-ACD (*Figures 3* and *4E*). A spin label at the aromatic region/conserved motif boundary (position 26) causes subtle peak intensity loss localized to loops L3/4 and L8/9 leading out of the interface groove, as would be expected if the conserved motif is bound in the groove and the aromatic region runs along one side of the ACD between the β4/β8 groove and the dimer interface groove (*Figure 5B*). Conserved motif binding at the dimer interface groove has been observed in HSPB6 and in HSPB2/3 hetero-tetramer crystals (*Clark et al., 2018*; *Sluchanko et al., 2017*). Given the high conservation of this motif and of residues that compose the dimer interface, it is likely that a similar relationship exists in HSPB1.

Although the Trp-rich peptide showed no detectable interactions with B1-ACD (*Figure 4F*), we were able to obtain some insights regarding its behavior. While attempting to introduce spin labels, we found the Trp-rich region to be highly sensitive to mutagenesis and were unable to probe its interactions through PRE experiments. Cysteine substitutions of S43 and V55 within the NTR-ACD construct yielded species larger than dimers and not amenable to NMR. An S50C NTR-ACD construct was dimeric but conjugation of MTSL at this position produced larger species. While the altered oligomeric propensity rendered these mutants unsuitable for PRE experiments, the ability of changes in this region to override the dimer-promoting mutations implicates the Trp-rich region in maintaining the delicate balance among oligomeric species. This sub-region harbors the disease-associated mutation P39L, which also affects HSPB1 oligomeric properties (see later section).

The insertion region was probed by a spin label at position 65, near the center of the region. Intermediate intensity loss was observed in most B1-ACD peaks, with some stronger effects in β9. The widespread but modest PREs suggest that the insertion region does not make sustained contact with any specific region of the ACD but rather 'hovers' over all faces of the ACD (*Figure 5C*). The region was omitted from peptide binding experiments due to its hydrophobicity and insolubility in aqueous buffer. Nevertheless, we were able to obtain backbone assignments for this region, which gave further insight into its behavior (discussed in next section).

Finally, the boundary region appears to interact at the dimer interface groove, as evidenced both by peptide binding and PRE results (*Figures 3*, *4G*, *5D and E*). Addition of the boundary region peptide (residues 74–91) to B1-ACD caused intensity loss and CSPs in peaks corresponding to residues near the dimer interface (*Figures 3* and *4G*). We were able to incorporate spin labels at two boundary region positions, 83 and 91, which flank the predicted β2 strand (residues 86–88 in PDB 4MJH). The spin label at position 91 (S91-SL) caused distinct intensity loss in peaks from loops L5/6 and L7/8 and from residues at the beginning of β3 and end of β9 (*Figures 3* and *5E*). The highly localized PREs on one side of the ACD are congruent with residue 91 being near the beginning of the β3 strand. In contrast, the spin label at position 83 caused wide-spread intensity loss in peaks corresponding to the dimer interface and the opposite side of the ACD (*Figures 3* and *5D*). In this

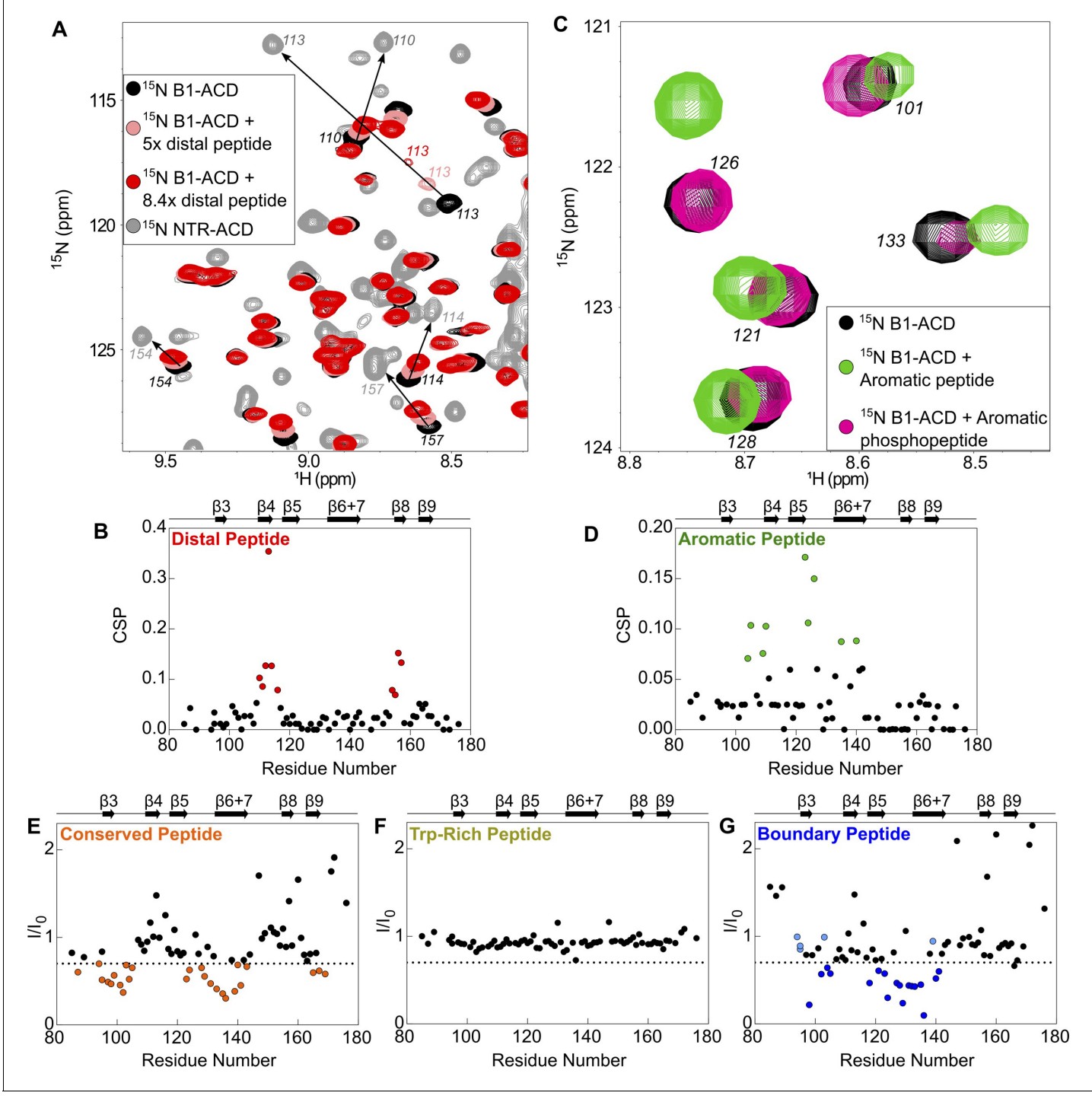

**Figure 4.** Perturbations of $^{15}$N-B1-ACD due to peptide binding. (A) The distal peptide, consisting of HSPB1 residues 1–13, causes CSPs in the $^{15}$N-HSQC spectrum of B1-ACD (black). Peak shifts occur along a trajectory toward the peak positions of the same residues in the $^{15}$N-HSQC spectrum of NTR-ACD (gray, NTR-ACD; pink, five molar equivalents; red, 8.4 molar equivalents). (B) The strongest CSPs (red dots) map to residues in the β4/β8 groove. (C) The aromatic peptide (residues 12–27) causes CSPs in the spectrum of B1-ACD (green vs. black), but these are weakened when the peptide contains phosphoserine at site 15 (pink). (D) The CSPs map to residues in loops 3/4 and 5/6 and strand β6+7 (green dots). (E) The conserved peptide (residues 25–37) causes intensity loss in peaks in the $^{15}$N-HSQC spectrum of B1-ACD corresponding to strands β3, β6+7, and β9. Peaks that lose more than 30% of their original intensity are colored in orange. (F) The Trp-rich peptide (residues 37–53) does not cause significant CSPs or intensity loss. (G) The boundary peptide (residues 74–91) causes both CSPs and intensity loss. Peaks that lose over 30% of their initial intensity are colored dark blue, and peaks with CSPs > 0.05 ppm that do not lose 30% of their initial intensity are colored light blue.

*Figure 4 continued on next page*

*Figure 4 continued*

DOI: https://doi.org/10.7554/eLife.50259.008

The following figure supplement is available for figure 4:

**Figure supplement 1.** Effect of phosphorylation on peptide binding.

DOI: https://doi.org/10.7554/eLife.50259.009

case, the ACD residues that are *not* affected by the spin-label are more informative: the data clearly show that position 83 does not approach loops L4/5, L7/8, and L9/CTR, all of which are localized on one side of the ACD dimer (the side affected by S91-SL). The fact that spin labels at positions flanking the putative β2 strand hit opposite sides of the ACD provides strong evidence that the boundary region spans the dimer interface groove in an antiparallel direction to strand β3, consistent with formation of a β2 strand. Whether this interaction is mutually exclusive with binding of the conserved motif to the dimer interface or can occur simultaneously cannot be deciphered from these experiments.

A boundary-region peptide that contains phospho-serine residues at positions 78 and 82 caused nearly identical perturbations to the non-phosphorylated peptide (*Figure 4—figure supplement 1B*), indicating that phosphorylation neither directly disrupts nor enhances the interaction between the boundary region and the dimer interface groove. However, the PRE results from S83-SL indicate that this region approaches residues that are perturbed by the aromatic peptide (*Figures 3* and *5D*), suggesting that the boundary region phosphorylation sites may be close in space to the S15 phosphorylation site, despite being over 60 residues apart in sequence.

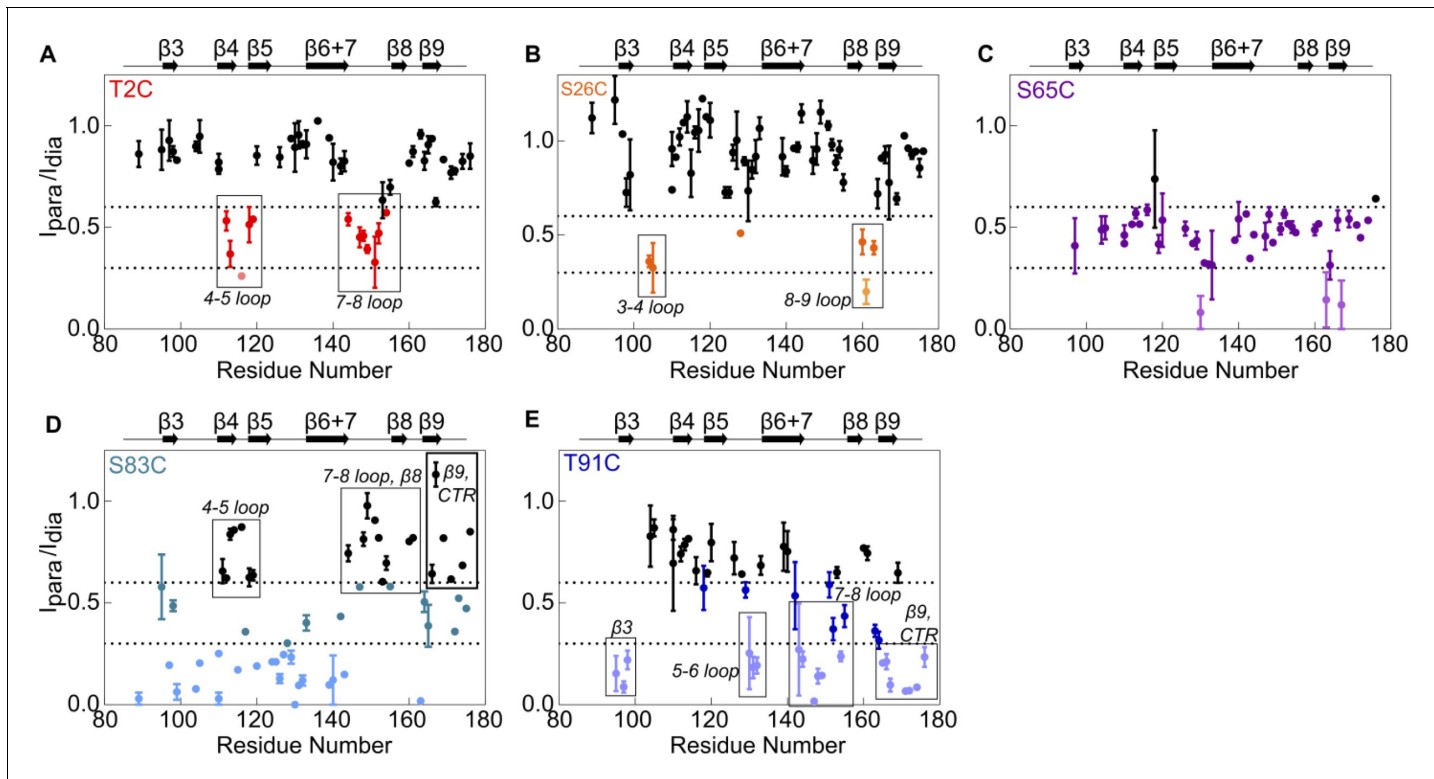

**Figure 5.** Paramagnetic relaxation enhancement NMR reveals contacts between NTR sites and residues in the ACD. The spin label MTSL was conjugated at five positions in the NTR of $^{15}$N-NTR-ACD. Spectra were collected in the presence of the active spin label (paramagnetic) and after it had been quenched by ascorbate (diamagnetic). Error bars represent the range of two independent experiments. Peaks that lose more than 40% of their intensity are highlighted in color, and peaks that lose more than 70% are shown in lighter colors.

DOI: https://doi.org/10.7554/eLife.50259.010

## Conformational heterogeneity is observed throughout the NTR

Peaks that are not assigned to ACD residues in the $^1$H-$^{15}$N HSQC spectrum of NTR-ACD presumably arise mainly from the NTR. Many of these peaks are weak (but reproducible across multiple protein preps) and were not observed in low-sensitivity 3D heteronuclear spectra despite high levels of deuteration and use of high field magnets equipped with cryoprobes. For resonances that were detected in heteronuclear spectra, there was considerable degeneracy in chemical shifts ($^{13}C_\alpha/^{13}C_\beta$), likely due to intrinsic disorder and/or conformational heterogeneity. While these properties hampered unambiguous assignment of NTR peaks, the weak and/or poorly-resolved peaks indicate that a majority of residues in the NTR exist in multiple environments, leading to their peak broadening. There are two clear exceptions: we were able to assign a contiguous stretch of residues that span the insertion and boundary regions, and several residues in the Trp-rich region.

Peaks corresponding to residues 62–79 could be assigned (*Figure 6A*); these span most of the (non-proline) residues of the insertion region and the start of the boundary region (*Figure 1C*). All peaks assigned for this stretch have high intensities (similar to CTR, *Figure 6C*) and chemical shifts consistent with random coil structure (*Figure 6A*). Only the N-terminal part of the boundary region is assigned, while the latter part of the boundary region that is predicted to form β2 could not be assigned. Notably, residues 64–69 were each assigned to two distinct peak sets, indicating two conformational states that interconvert slowly, despite their both having 'random coil' chemical shifts. Based on the observed chemical shift differences and the slow exchange condition, we can estimate an upper limit for the exchange rate of ~25 s$^{-1}$. The second conformation represents a substantial fraction of the population, as judged from relative peak intensities (>20%). A short contiguous stretch at the C-terminal end of the Trp-rich region (residues 46–49) was assigned, with Ser49 having two peak sets. Thus, both regions exist in at least two distinct conformations and/or environments. The observed behavior may arise from *cis* and *trans* conformations of one or more of the numerous proline residues in the insertion and Trp-rich regions. However, $^{13}$C chemical shifts needed to confirm proline isomerization could not be obtained from the data and we did not pursue mutational analysis to identify the source(s) of heterogeneity due to the abundant prolines in the regions.

Assignments of NTR HSQC peaks provided information regarding proximities of NTR regions to each other from PREs (*Figure 6B*). The spin label at insertion region residue 65 yields strong PREs across the assigned NTR peaks, validating their assignments to this region of the sequence. The position two spin label did not yield PREs in assigned NTR residues or in weak unassigned peaks, indicating that the distal region's locations do not overlap with those of the Trp-rich or insertion regions. A position 26 spin label at the beginning of the conserved region gives modest-to-strong PREs to the Trp-rich and boundary regions but weak PREs to the insertion region. The position 83 spin label gave strong PREs to the boundary and insertion regions, and the position 91 spin label gave moderate PREs to the same regions. Together, this provides evidence for extensive interactions between the sub-regions of the NTR. In particular, the conserved and Trp-rich regions seem to be in close contact with each other, as are the insertion and boundary regions. Only the distal region appears not to make extensive contact with other parts of the NTR. The data support a model in which the NTR exists in a compact state with extensive NTR/ACD and NTR/NTR contacts as well as some residual structure, rather than a more flexible, extended conformation seen for many intrinsically disordered regions.

To investigate the dynamics of assigned NTR residues, we measured transverse relaxation rates (R2) of peaks in the NTR-ACD spectrum (*Figure 6D*). Residues in the ACD have R2 values of ~40 s$^{-1}$, consistent with their positions within an ordered region. NTR insertion sub-region residues and the final (unstructured) residues in the ACD have lower R2 values (around 10 s$^{-1}$), consistent with higher flexibility. The limited number of other assigned NTR resonances have R2 values that range from around 20 s$^{-1}$ to ~60 s$^{-1}$, consistent with greater conformational restriction of these NTR regions and/or exchange between multiple states on the millisecond timescale.

We used HDXMS to compare NTR properties in dimeric and oligomeric forms of HSPB1. Backbone amide protons in structured and/or buried regions exchange with deuterons more slowly (i.e., are 'protected' from exchange) than those in unstructured or accessible regions. HDXMS can be performed on proteins of any size under a variety of solution conditions, providing an opportunity to obtain peptide-level information on the dynamics of HSPB1 oligomers. Time courses of deuterium exchange from three seconds to one hour were measured at room temperature for C137S-HSPB1

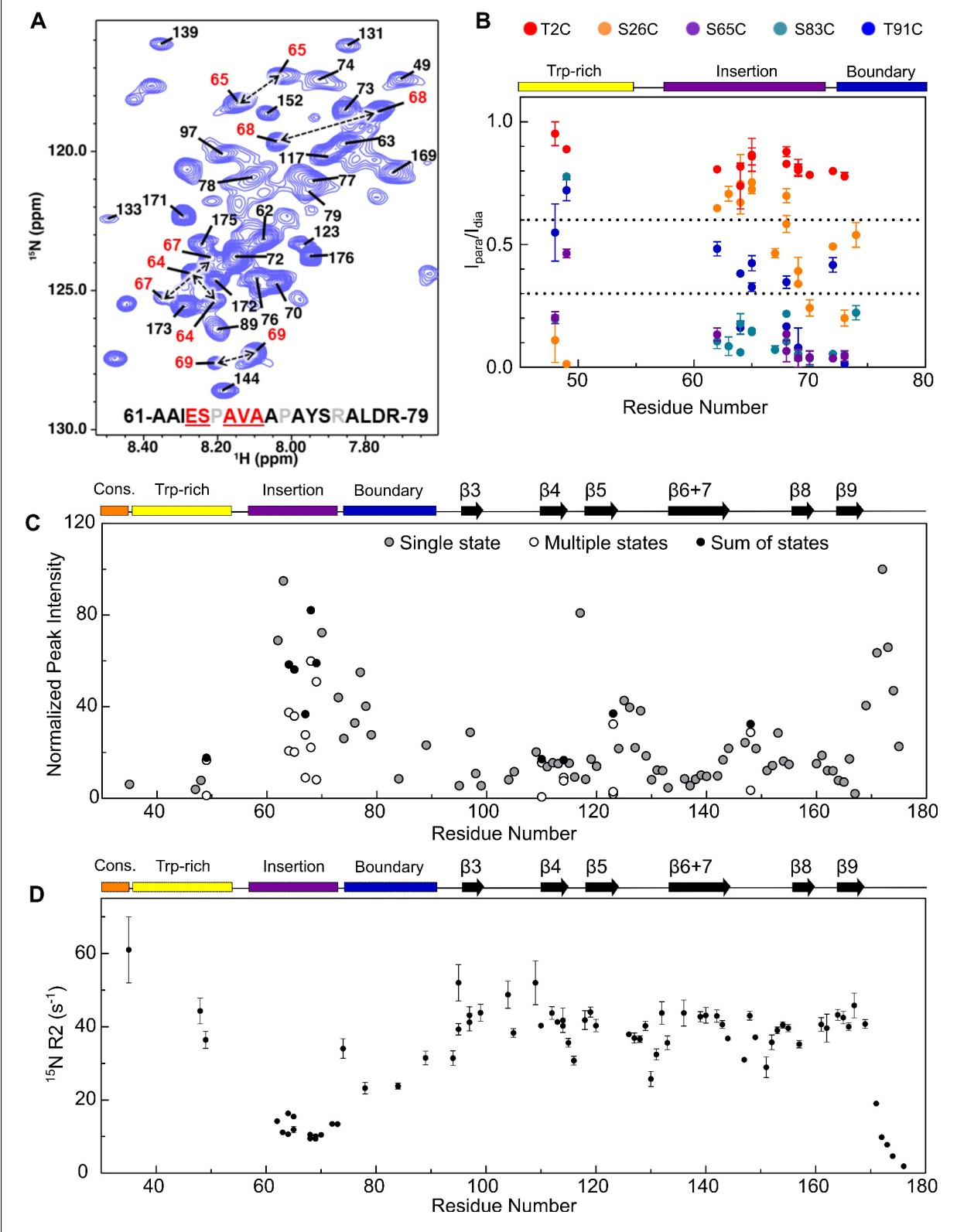

**Figure 6.** Assignment of NTR residues reveal disorder and heterogeneity. (**A**) $^1$H-$^{15}$N HSQC-TROSY spectrum of NTR-ACD, highlighting strong peaks in the center of the spectrum that correspond to disordered regions of the protein (NTR and CTR). For part of the NTR, pairs of peaks are assigned to the same residue indicating conformational heterogeneity (red in sequence and peak labels). (**B**) Summary of PRE effects from previously described spin

*Figure 6 continued on next page*

*Figure 6 continued*

label positions on assigned regions of the NTR. (C) Relative intensities of all non-overlapping peaks assigned in NTR-ACD, including sums of peaks corresponding to different conformations (most intense peak = 100). (D) $^{15}N$ transverse relaxation rates (R2) of assigned residues in NTR-ACD.

DOI: https://doi.org/10.7554/eLife.50259.011

oligomers and C137S-HSPB1$_{dimer}$. The sole Cys residue, C137, that resides at the dimer interface was substituted to abrogate the need for reducing agent to inhibit disulfide bond formation across the dimer interface that could confound the analysis. Pepsin digestion yielded peptides across the full length of the protein (peptide statistics are shown in *Figure 7—source data 2*).

Deuterium uptake levels differed most dramatically across regions of HSPB1 in the earliest (3 s) timepoint, shown in *Figure 7* (see *Figure 7—source data 1* for full kinetic information). The profile for HSPB1$_{dimer}$ (middle panel, *Figure 7*) is congruent with observations and conclusions drawn from NMR presented above. Residues 87–172, which span the ACD, show protection even at the latest time point, which is consistent with its β-sandwich structure. Peptides within the aromatic, insert, and boundary regions in the NTR, along with the far C-terminal region (residues 187–205) of the CTR are all highly deuterated within 3 seconds, consistent with their lacking stable secondary structure. Interestingly, peptides along the distal, conserved, and Trp-rich regions of the NTR showed moderate protection, indicating the presence of some local structure. All peptides covering the distal region displayed a bimodal isotopic mass envelope at the early time points indicating that there are two distinct populations of the distal region: one that exchanges readily with deuterium and one that is protected (see *Figure 7—figure supplement 1*). The bimodal distributions observed for these peptides may reflect the populations of the NTR that are either bound and sequestered or free and solvent-accessible. A substantial protected population is present through 15 seconds, implying that the lifetime of the bound state is on the order of at least several seconds. All the other peptides outside of the distal region displayed only unimodal spectra in the HSPB1$_{dimer}$ dataset.

The general features of the exchange profile for oligomeric HSPB1 (top panel, *Figure 7*) are similar to HSPB1$_{dimer}$, with the ACD being most protected and the NTR and CTR less protected. As in the dimer, the insertion region, boundary region, and CTR have the highest rate of deuterium uptake, indicating that they remain unstructured and accessible in oligomers. However, the aromatic region shows more protection in oligomers than in HSPB1$_{dimer}$, indicating some involvement in the oligomerization. The most striking feature of the oligomeric HSPB1 HDX profile is the large number of bimodal peptides that are observed. The distal region showed similar bimodal spectra as seen with the HSPB1$_{dimer}$, except the relative population of the more protected species was diminished (~40% vs. 80%). Beyond the distal peptides, another six peptides have two distinguishable populations in the context of HSPB1 oligomers. Peptides that arise from the conserved, Trp-rich, and boundary regions in the NTR display two populations. The conserved and boundary regions were observed to interact with the ACD in the dimer context from NMR results, so either these interactions are longer-lived within the confines of oligomers, or they may also be directly involved in oligomeric interactions. The presence of a substantial protected population at 4 minutes indicates a lifetime of interaction of several minutes in the oligomeric context. In the CTR, a peptide that contains the IXI motif (mutated to GXG in HSPB1$_{dimer}$ to inhibit its binding to the β4/β8 groove) has a population (~45%) that is protected from deuterium exchange and one that is completely exchanged. This behavior is consistent with the notion that IXI-containing CTRs exist in both free and β4/β8-bound states.

The populations observed for the bimodal spectra are not sufficiently resolved along multiple timepoints to determine precise rates of slow conformational exchange ('EX1 kinetics') or to make quantitative comparisons. Nevertheless, qualitative comparisons at the well-resolved 3 s timepoint can be made between regions or between protein constructs. The weighted average deuteration for each peptide with bimodal exchange in *Figure 7* provides a qualitative estimate of the relative populations of 'bound' and 'free' states. For example, both the distal and IXI-CTR peptides show a substantial proportion (~40–45%) of the region in a protected state. These two regions bind to the same ACD groove, creating a situation where there are twice as many potential β4/β8 groove-binding sequences as there are grooves. The HDX data indicate that enough groove-binding sequences

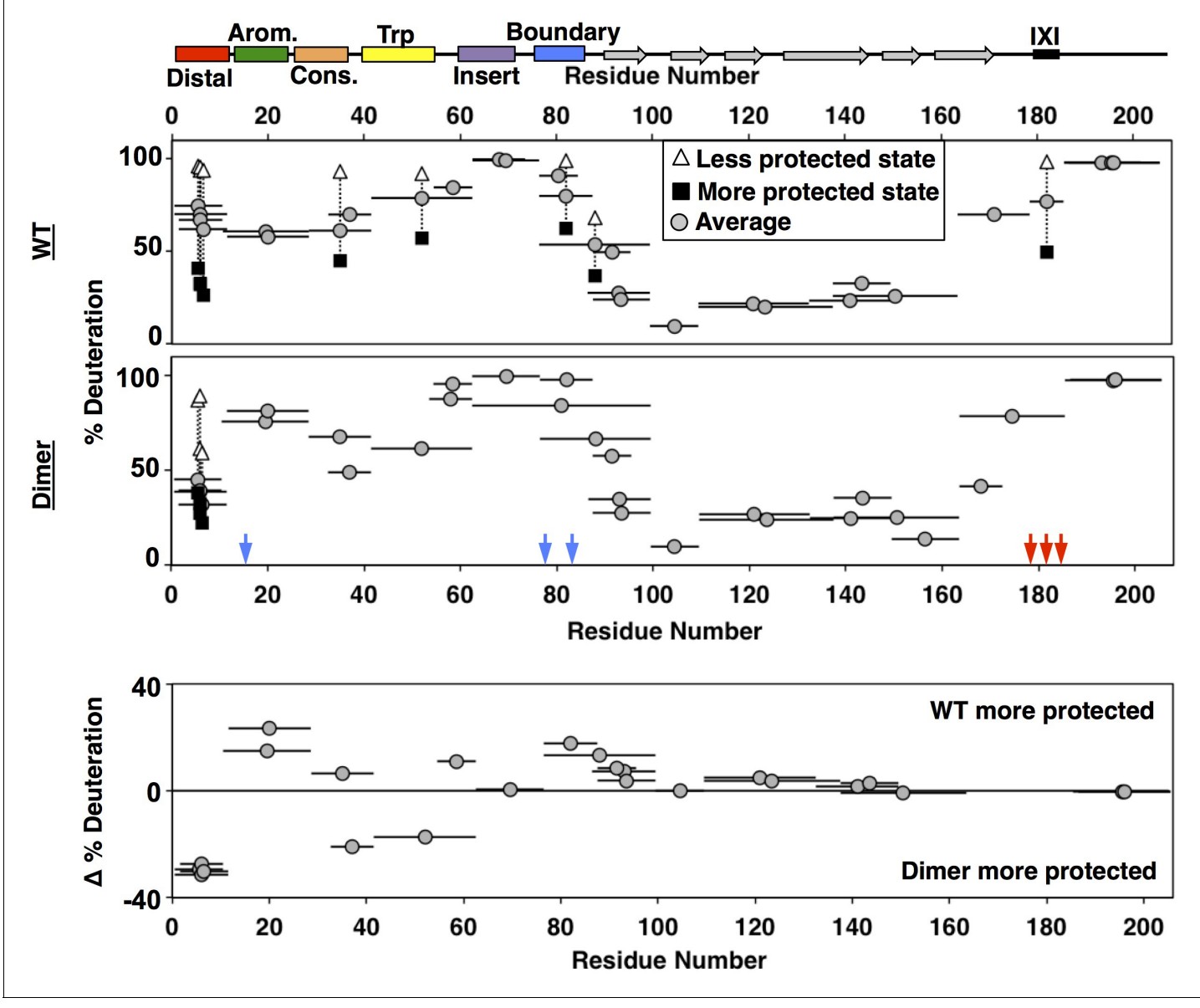

**Figure 7.** HDXMS analysis of WT oligomers and HSPB1$_{dimer}$ reveals changes in protection and heterogeneity in the NTR. Representative peptides are indicated as horizontal bars. The midpoint of each peptide is represented by a gray circle showing the deuteration level of the peptide after 3 s (see Source Data one for full kinetic table). For peptides that show a bimodal distribution (different states), black squares and white triangles represent the more- and less-deuterated populations, and the gray circle represents the weighted average deuteration. Blue and red arrows indicate sites of mutation used to generate HSPB1$_{dimer}$. The difference in deuteration level between WT and HSPB1$_{dimer}$ is shown for each peptide in the bottom plot. Although identical peptides for the start of the CTR cannot be compared due to the mutations introduced, the profile for the 164–185 peptide in HSPB1$_{dimer}$ is consistent with the high level of deuteration observed for peptides 164–178 and 179–185 in WT oligomers.

DOI: https://doi.org/10.7554/eLife.50259.012

The following source data and figure supplements are available for figure 7:

**Source data 1.** HDXMS profiles for all forms of HSPB1.
DOI: https://doi.org/10.7554/eLife.50259.015

**Source data 2.** Statistics for HDXMS experiments on all mutants.
DOI: https://doi.org/10.7554/eLife.50259.016

**Figure supplement 1.** Example bimodal spectra for peptides 1–10, 29–41, 77–99, and 179–185 of C137S oligomers.
DOI: https://doi.org/10.7554/eLife.50259.013

**Figure supplement 2.** Comparison of changes in HDXMS between WT and HSPB1$_{dimer}$ across biological replicates at 3 s.
DOI: https://doi.org/10.7554/eLife.50259.014

are solvent-protected to imply that the majority of β4/β8 grooves are occupied in HSPB1 oligomers and that, on average, half the distal sub-regions and half the CTR-IXIs are bound.

To get a more complete overall picture of what else is different in the oligomeric form, we compared the deuteration levels for each identical peptide that could be identified between HSPB1$_{dimer}$ and HSPB1 oligomers, using the weighted average percent deuteration for bimodal peptides. In the bottom panel of *Figure 7*, peptides that appear above the 0% line are more protected in oligomers and peptides that fall below the 0% lines are more protected in dimers. Overall, the ACD displays similar deuterium uptake regardless of whether it is in an isolated dimer or a large oligomer. This indicates that the extent of deuterium exchange at this timepoint is predominantly dictated by the β-sandwich structure. All peptides arising from the distal region show more protection in HSPB1$_{dimer}$, with the weighted deuteration skewed to the more protected state (larger proportion) consistent with a larger fraction of the N-terminus bound in the dimer. This may be due to the CTR mutations introduced to generate the dimer, as the lack of IXI motif in the CTR removes it from competition for the β4/β8 groove. Two other contiguous peptides (33–41 and 42–62) that span conserved and Trp-rich regions show greater protection in the dimer. The overlapping 29–41 peptide is more protected in the oligomer, indicating that protection occurs in the first few residues of the peptide in oligomers, whereas protection occurs in later residues in the dimer, consistent with distinct interactions or structural features. The NMR results identified an interaction between the conserved region and the ACD dimer interface, but provided no evidence for an interaction involving the Trp-rich region and the ACD. According to secondary structure predictions, the Trp-rich region has the highest propensity of any NTR region to adopt secondary structure. Intriguingly, the CD spectrum of HSPB1$_{dimer}$, but not HSPB1 oligomer, has an unusual positive peak at 230 nm (*Baughman et al., 2018*). Similar features have been attributed to exciton couplets that can arise from interactions between Trp rings. Together, the HDX and CD data suggest that the Trp-rich region adopts specific structure in dimeric species that is not populated to a detectable degree in the oligomer. The two regions that are less protected in HSPB1$_{dimer}$ are the aromatic and boundary regions, which contain the phosphorylation-mimicking mutations. The NMR data indicate that the conserved and boundary regions both interact with the dimer interface groove. The increased accessibility of the aromatic region is consistent with the reduced binding of the phosphorylated form of the aromatic peptide. It is possible that release of the aromatic region, which neighbors the conserved region, is coupled to decreased protection of the boundary region. Formation of a new structural feature involving the Trp-rich region might only be possible when neighboring regions are released from their ACD contacts.

## Modeling inter-region interactions in HSPB1$_{dimer}$

The results presented above indicate that the HSPB1 NTR contains distinct regions that reside in specific locations relative to the well-defined ACD dimer and that, in many cases, make direct contact with the ACD. We sought to combine the information garnered from the peptide-binding, PRE, and HDXMS experiments into a set of structural models. Our goals for this modeling process were two-fold. First, the models aid in visualization of the NTR-ACD interactions described above. Second, the modeling process allows us to determine which combinations of NTR-ACD interactions can generate physically realistic structural models, and thus whether any combination of NTR-ACD interactions may not be physically possible. These structures are intended neither as a complete sampling of HSPB1 conformational space nor as atomic-level models of specific interactions.

We produced a homology model by combining the crystal structure of the HSPB1 ACD (PDB 4MJH) with peptides from the crystal structure of the HSPB2/3 hetero-tetramer (PDB 6F2R) using the PyMOL Molecular Graphics System. The starting model included all of the possible NTR-ACD interactions identified above: two copies of the β2 strand, a distal motif bound in each β4/β8 groove, and a copy of the conserved motif bound in the dimer interface groove. Clashes between the HSPB1 ACD structure and the peptides from the HSPB2/3 crystal structure were eliminated and missing loops were modeled in using PyRosetta. In total, there are four possible ways to connect the fragments in the homology model: either copy of the distal motif in the β4/β8 grooves could be connected to the conserved motif at the dimer interface, and the conserved motif could be connected to either copy of the β2 strand (*Figure 8A*). In total, we created four ensembles of 100 structural models, which each sampled a different way of connecting the NTR peptide fragments in our initial homology model (*Figure 8B, C* and *Figure 8—figure supplement 1*). We found that it

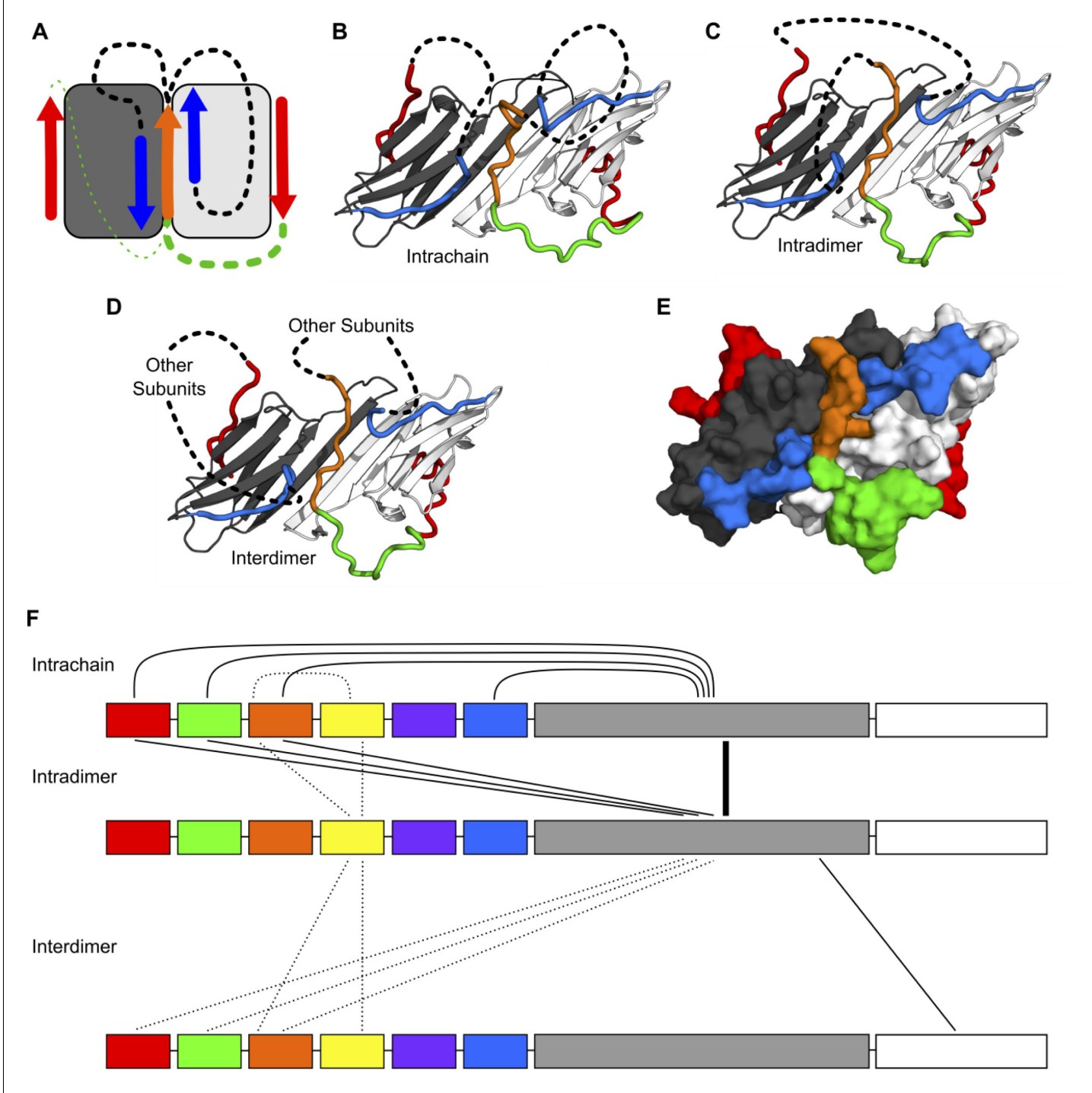

**Figure 8.** Modeling of NTR-ACD interactions. (**A**) Cartoon showing the starting structure of the ACD with two copies of the distal motif (red arrows), one copy of the conserved motif (orange arrow), and two copies of β2 (blue arrows). The missing loops can be modeled with four potential connectivities. Either copy of the distal motif can be connected to the conserved motif (green lines), which can then be connected to either copy of β2 (black lines). Based on our NMR results, we believe it more likely that the conserved motif is connected to the distal motif oriented in the opposite direction (thicker green line), so we only include structures with this connection in this figure. (**B**) If the conserved motif is connected to the β2 strand oriented antiparallel to it, the contacts between the distal and aromatic regions and the ACD occur within the same polypeptide chain. (**C**) If the conserved motif is connected to the other β2 strand, the contacts occur between different chains but within the same dimer. (**D**) It is likely that within the context of a higher-order oligomer, similar contacts could occur between subunits that are not part of the same ACD dimer building block. (**E**) A surface representation of the model in panel C shows that the regions of the NTR included in our model make extensive contact with the ACD and

*Figure 8 continued on next page*

*Figure 8 continued*

match the perturbed regions highlighted in *Figure 3*. (F) The types of intra- and interchain contacts that are possible within HSPB1 dimers and higher-order oligomers are outlined. Solid lines represent interactions for which we have evidence from our NMR, HDXMS, and modeling data. Dotted lines represent hypothetical interactions for which we do not have direct evidence but which we believe are likely to occur.

DOI: https://doi.org/10.7554/eLife.50259.017

The following figure supplement is available for figure 8:

**Figure supplement 1.** Models in which the conserved motif is connected to the distal motif oriented in the opposite direction.

DOI: https://doi.org/10.7554/eLife.50259.018

was possible to develop realistic structural models that contain all types of NTR-ACD interactions identified above, as well as all possible loop connectivities. The dimer interface groove is able to accommodate two copies of the β2 strand in addition to the conserved motif, and both β4/β8 grooves are able to bind a copy of the distal motif. These peptide fragments can be connected to each other in all four conceivable ways.

If the conserved motif is connected to the distal motif oriented in the opposite direction, the aromatic region forms a loop along the side of the ACD containing loops 3/4, 5/6, and 8/9 to connect these regions (*Figure 8B and C*). Given our peptide-binding NMR results for the aromatic region, we believe that this configuration is favored, particularly in the non-phosphorylated state. Nevertheless, it is also physically possible for the aromatic region to connect the conserved motif to the distal region in the opposite groove, in which case it spans across the top of the ACD β-sandwich, contacting the β8, β9, and β3 strands (*Figure 8—figure supplement 1*). While we do not observe experimental evidence for this interaction, we cannot rule it out, particularly in the phosphorylated state in which the aromatic region has low affinity for the ACD. In either case, the conserved motif can then be connected to either one of the β2 strands, while the distal motif not connected to the bound conserved motif can be connected to the other. These connections consist of ~50–75 residues and contain part of the boundary region, the insertion region, the Trp-rich region, and (for the chain with an unbound conserved motif) the conserved and aromatic regions. Given their length, they can adopt multiple conformations and orientations relative to the ACD. We have omitted these regions from the models shown in *Figure 8* and its supplement for clarity and to avoid over-interpretation of the structural aspects of regions for which we have limited experimental data. Surface representations of these models show that the locations of the NTR sub-regions are in good agreement with the ACD surfaces perturbed in the peptide binding and PRE NMR experiments (comparing *Figure 3* with *Figure 8E*).

Overall, the results from the modeling suggest that any combination of the NTR-ACD interactions defined in our study is physically feasible. While we only created models containing the maximum NTR-ACD interactions supported by our experimental data, any of the interacting motifs we have modeled could dissociate from the ACD and adopt a more disordered conformation. The results from our NMR and HDXMS experiments indicate that most of these NTR regions occupy both ACD-bound and ACD-unbound conformations, so it is likely that multiple combinations of NTR/ACD interactions occur in solution. Additionally, the similarity of protected regions in the HDXMS profiles of HSPB1 dimers and oligomers indicate that the interactions depicted in these models also occur within higher-order oligomers. The peptide fragments depicted in our dimeric models could conceivably be connected to other ACD dimers or monomers within an oligomer (*Figure 8D*). The array of possible interactions within sHSP oligomers is depicted in *Figure 8F*. Many regions can form intra-chain, intra-dimer, and inter-dimer interactions. The possibility for multiple combinations of interactions and connectivities contributes to the high degree of plasticity and heterogeneity observed for HSPB1 in NMR and HDXMS experiments.

## Disease mutations in the NTR have differential effects on HSPB1 structure

To leverage new insights regarding local structure in the NTR and interactions with the ACD, we sought to assess how reported disease-associated mutations in the HSPB1 NTR affect structure and/or dynamics. We chose two NTR disease mutants whose effects on oligomer size (larger than WT) and chaperone function have been previously characterized: G34R and P39L mutants in the

conserved and Trp-rich regions, respectively (*Muranova et al., 2015*). To identify localized effects of the mutations, we performed peptide binding experiments with mutant peptides and B1-ACD. To detect potential global effects, we performed HDXMS analysis on mutant proteins. Glycine at position 34 is the final conserved residue in the 'conserved motif' found among orthologs and paralogs; its mutation to Arg is expected to reduce flexibility of the backbone and/or alter the region's interactions. At the start of the Trp-rich region, Pro39 is conserved among orthologs but not among paralogs. Notably, Pro39 is two residues before a region that is predicted to have helical propensity (residues 41–46, *Figure 1—figure supplement 1*). Mutation of Pro39 to leucine might change the structural propensity of this region by increasing flexibility and/or favoring helical formation. As presented below, we found highly divergent local structural effects of the two mutations, despite their close proximity in sequence.

As presented above, the conserved motif peptide (residues 25–37) causes peak broadening in dimer interface groove residues (i.e., β3 and β6+7) in $^{15}$N-B1-ACD spectra. An otherwise identical peptide that contains the G34R substitution yields greatly reduced perturbations (*Figure 9B*), indicating that the bulky, charged Arg sidechain disrupts the ability of the conserved region to interact with the dimer interface groove. Consistent with the notion that the environment of the mutated conserved region is altered, increased deuterium uptake for the fragment that contains the G34R substitution in otherwise wild-type HSPB1 (oligomer) is observed by HDXMS (*Figure 9A*, top and bottom panels). The increase in deuterium uptake was sufficiently large (as high as 25% in early time points), that it likely reflects a true increase in local flexibility, rather than a consequence of the G to R substitution altering the intrinsic exchange rate of this peptide (*Wales et al., 2016*). Unexpectedly, the NTR boundary region also shows enhanced deuterium uptake in G34R-HSPB1 and, strikingly, no longer shows bimodal behavior. All other portions of oligomeric HSPB1 are not significantly changed in their exchange profiles.

The non-local effect of the G34R mutation on the boundary region suggested to us that there could be coupling between the two NTR regions, similar to possible coupling between phosphorylation sites in the aromatic and boundary regions mentioned earlier. We took advantage of another identified disease-associated mutation that similarly substitutes arginine for glycine, but in the boundary region (G84R) to test this hypothesis. Intriguingly, HDXMS data on the mutant (*Figure 9A*) revealed enhanced deuterium uptake in both the boundary region and in the conserved region, confirming that the dispositions of these two non-contiguous NTR regions are inter-dependent. The NMR data indicate that both regions interact with the ACD dimer interface-groove, and our modeling suggests that both interactions can occur simultaneously. While the NMR data do not address whether the binding of one influences the other, the HDXMS data provide some clarity on this question. The observed inter-dependence of protection from deuterium exchange implies that each region enhances the ability of the other to contact the groove and that the conformation that leads to protection in the HDXMS experiment has both a conserved region and a boundary region present in the groove. Whether the two bound sub-regions are on the same HSPB1 polypeptide or different ones within a dimer or oligomer cannot be ascertained from these experiments, but our modeling implies that both cases are likely to occur.

As mentioned above, we did not detect evidence of binding for the Trp-rich region peptide to $^{15}$N-B1-ACD. We obtained similar results for a peptide that contains the P39L substitution, indicating that the mutation does not lead to a gain of binding function (*Figure 9C*). Nevertheless, introduction of P39L into otherwise wild-type HSPB1 has a marked effect on the HDXMS profile (*Figure 9A*). The first four NTR sub-regions are substantially more protected from deuteration in mutant oligomers, while the ACD is essentially unaffected. The largest increase in protection is observed in the fragment that contains the mutation (29-41). The comparison to WT is complicated by the addition of an extra amide from the P39L mutation, but the massive decrease in exchange (25%) is consistent with the predicted increased helical propensity in this region and an observed increase in helicity revealed in the CD spectrum (*Figure 9—figure supplement 1*). Altogether, the observations suggest that the P39L mutation increases local secondary structure and/or promotes NTR-NTR contacts. In further support of this conclusion, the subunit exchange rate for P39L-HSPB1 is three-fold slower than for either wild-type or G34R oligomers (*Figure 9—figure supplement 1*).

Altogether, the results highlight a variety of alterations that occur in the HSPB1 NTR under situations such as phosphorylation or disease mutation. These alterations can have profound non-local

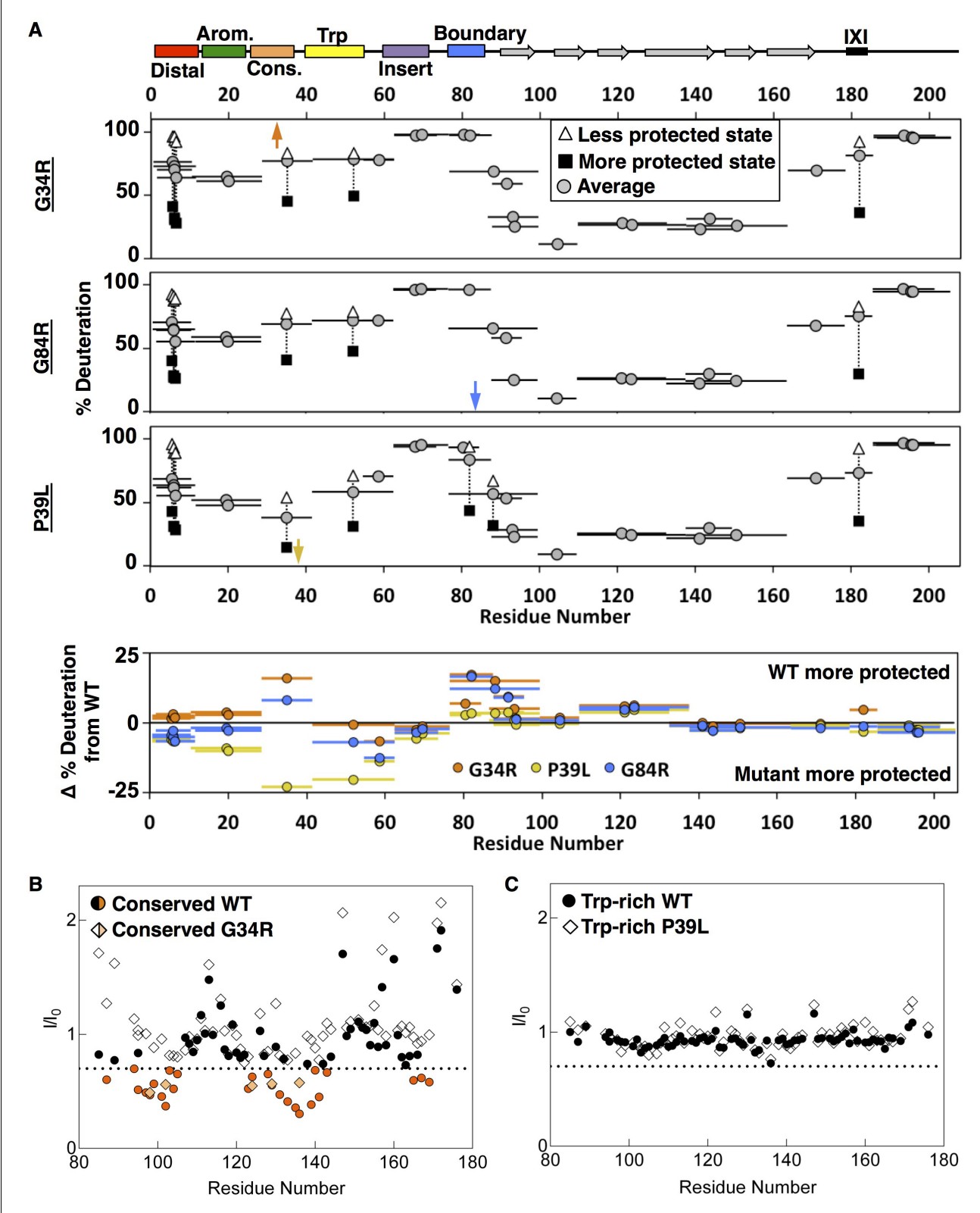

**Figure 9.** Analysis of disease-associated HSPB1 NTR mutations G34R, G84R, and P39L. (**A**) HDXMS analysis of each mutant. Representative peptides are indicated as horizontal bars. The midpoint of each peptide is represented by a gray circle showing the deuteration level of the peptide after 3 s (see Source Data one for full kinetic table). For peptides that show a bimodal distribution (different states), black squares and white triangles represent the more or less deuterated populations, and the gray circle represents the weighted average deuteration. Arrows indicate sites of mutation (G34R,

*Figure 9 continued on next page*

*Figure 9 continued*

G84R, and P39L). The deuteration difference from WT for each disease mutant is shown for each peptide in the bottom plot. (**B**) NMR intensity loss in G34R-modified conserved peptide binding experiment with ACD. 'WT' conserved peptide intensity ratios are shown as circles, and G34R peptide ratios as diamonds. Affected residues below the dotted line are colored orange. Fewer residues are affected by mutant peptide binding and to a lesser extent. (**C**) Analogous peptide binding experiment with Trp-rich peptide and P39L-modified peptide. In both cases, no notable intensity losses occur.

DOI: https://doi.org/10.7554/eLife.50259.019

The following figure supplement is available for figure 9:

**Figure supplement 1.** Circular dichroism spectra and subunit exchange kinetics of disease mutants.

DOI: https://doi.org/10.7554/eLife.50259.020

effects on other sub-regions of the NTR, leading to global changes in HSPB1 structure and oligomerization.

## Discussion

Despite being among the most ubiquitously expressed of the ten human sHSPs, there is a paucity of structural information regarding HSPB1. As for all known examples, the central ACDs of two subunits of HSPB1 adopt a dimeric β-sandwich structure (*Rajagopal et al., 2015a*; *Baranova et al., 2011*; *Rajagopal et al., 2015b*; *Hochberg et al., 2014*; *Clark et al., 2018*; *Sluchanko et al., 2017*). However, there has been a complete lack of structural information for the remaining ~50% of HSPB1, most of which is represented by the enigmatic NTR. To overcome the challenges posed by high heterogeneity and polydispersity of large HSPB1 oligomers, we sought to obtain new structural information from a more tractable dimeric form of HSPB1 that retains its chaperone activity. Despite its monodispersity in solution, HSPB1$_{dimer}$ exhibits substantial heterogeneity as detected by both NMR and HDXMS. Exemplary of the heterogeneity and dynamics, a majority of the NMR resonances from NTR residues are either of low intensity or are broadened beyond detection even at high magnetic fields, signifying multiple environments for these residues. The disordered NTRs of sHSPs have remained enigmatic for decades, with very little structural information emerging. The hydrophobic NTRs are required for oligomerization of some sHSPs (HSPB1, HSPB4, HSPB5), although paradoxically others remain predominantly dimeric despite having similarly long and hydrophobic NTRs (HSPB6 and HSPB8). Our effort to generate a well-behaved dimer of HSPB1 and characterize its NTR both when tethered to its ACD and when presented as short peptides that represent NTR sub-regions was surprisingly revealing. Peptides representing sub-regions of the NTR display specific interactions with the ACD of HSPB1. Furthermore, PRE experiments revealed that the interactions have preferred orientations. In addition to the previously observed interaction of the IXI motif from the CTR with the β4/β8 groove, our results establish four distinct NTR/ACD interactions: 1) distal region with β4/β8 groove, 2) aromatic region with L3/4 and L5/6, 3) conserved region with dimer interface groove, and 4) boundary region with dimer interface groove (*Figure 8F*). The fact that multiple HSPB1 regions can bind to a given groove or surface sets up a situation in which there are more potential binding elements than there are binding sites. This, in turn, creates a large combinatorial array of possible states within a dimer, and even more states within an oligomer. For example, each HSPB1 dimer contains two β4/β8 grooves and four interacting regions (two copies each of the distal region and the CTR IXI motif). Our data and modeling indicate that a given groove may be 1) empty, 2) bound by an inter-chain distal region, 3) bound by an intra-chain distal region, or 4) bound by a CTR IXI motif (usually from another dimer within an oligomer). A dimer may have zero, one, or two of its grooves filled, presumably with any combination of binders. The HDXMS results on HSPB1 oligomers indicate that a large proportion (roughly half) of the distal regions and the CTRs are bound, meaning the β4/β8 grooves must be predominantly occupied. Each HSPB1 dimer has a single dimer interface groove, but its potential interactions with two NTR regions creates a similarly complicated situation: a given dimer interface groove may be empty, bound by a single boundary region, a single conserved region, two boundary regions, one boundary plus one conserved, or two boundary regions plus a conserved region. Again, in the context of an oligomer, the combinatorial possibilities will be increased if the interactions can occur from neighboring dimer units.

While the NTR of sHSPs has long been known to be intrinsically disordered, it is clear from this work and recent crystal structures that further definition is required to accurately describe and define

sHSP NTRs. Indeed, the only segment of the NTR that behaves like an ideal, well-solvated random coil is the insertion sub-region. In recent years, it has come to be appreciated that the term intrinsic disorder can encompass a broad variety of behaviors distinct from the random coil limit. Disordered proteins or regions that adopt compact conformations and that transiently sample secondary structure have been described as globule-like, and disordered regions that interact with binding partners in a dynamic manner have been described as fuzzy (*Dyson and Wright, 2005*; *van der Lee et al., 2014*). By definition, fuzzy protein-protein interactions cannot be described by a single conformational state (*Tompa and Fuxreiter, 2008*). However, given the high degree of orientational specificity of many NTR-ACD interactions, these interactions can be described neither as fuzzy in the canonical sense, nor as molten globule-like. The only region of the NTR that could be said to interact with the ACD in a fuzzy manner is the insertion region, as a spin label placed in this position causes non-specific PREs in many regions of the ACD, inconsistent with a single conformational or orientational state.

Notably, ordered interactions occur for several NTR sub-regions with the ACD with varying levels of affinity and some interactions appear to be interdependent. The high degree of heterogeneity in HSPB1 dimers and oligomers is generated not by multiple random or fuzzy states but rather by the large number of possible combinations of several specific and orientationally-defined states. Based on observation of multiple slowly exchanging peaks by NMR for certain residues, elevated transverse relaxation rates, and bimodal HDXMS at long time points, the lifetimes for these interactions range from a minimum of tens of milliseconds to several minutes. For this reason, we propose the term 'quasi-ordered' to describe the NTR of HSPB1, as it makes specific long-lived (on the timescale of seconds) contacts while remaining dynamic and heterogeneous. In our definition, quasi-ordered interactions sample an ensemble of well-defined, relatively long-lived, approximately isoenergetic states; they are more ordered than fuzzy complexes yet more heterogeneous than systems that display folding-upon-binding (*Dyson and Wright, 2005*).

The structural information gleaned from the approaches presented here on HSPB1 add substantially to emergent structural insights on human sHSPs. First, knob-into-hole binding of CTR IXI motifs and β4/β8 grooves has been well established in numerous sHSPs. This type of interaction was recently reported for NTR IXI motifs as well (*Clark et al., 2018*; *Sluchanko et al., 2017*). Recognition of non-canonical hydrophobic motifs by the β4/β8 groove has been shown previously for the client proteins amyloid-β and α-synuclein and in crystal structures of artificially-truncated sHSP constructs (*Collier et al., 2019*; *Mainz et al., 2015*; *Liu et al., 2018*; *Weeks et al., 2018*; *Weeks et al., 2014*), but never within a full-length sHSP. Our results show that a motif of alternating hydrophobic residues in the HSPB1 distal region competes effectively with the canonical IXI motif in the protein's CTR, expanding the repertoire of potential binding partners.

The eight-residue conserved motif is the only stretch of identifiable sequence conservation in the NTR of human sHSPs. Recently, conserved motifs bound at the dimer interface groove have been observed in HSPB6 and HSPB2/3 structures. In the HSPB6 structure, the conserved sequence occupies the dimer interface groove and no β2 strand is present (*Sluchanko et al., 2017*). In the HSPB2/ HSPB3 structure, the conserved motif of HSPB2 and one copy of a β2 strand from the same protomer occupy the groove. Our modeling shows that it is physically possible for two copies of β2 strand and a conserved motif to be bound simultaneously. The reciprocal effects observed in the HDXMS of disease-associated mutations in these two regions (G34R- and G84R-HSPB1) strongly suggest that disruption of one of the interactions destabilizes the other.

An interaction analogous to the one defined for the aromatic sub-region of HSPB1 has not been previously reported. Our results indicate that the interaction is favored in the absence of Ser15 phosphorylation, which disrupts the interaction. Only two other human sHSPs are enriched in aromatic residues in this region, namely HSPB4 and HSPB5. HSPB1, B4, and B5 are also the only human sHSPs known to form large oligomers, leading us to propose that the aromatic region/ACD interaction may be a driver of large oligomer formation. Notably, HSPB1 and HSPB5 are both phosphorylated in response to stress conditions on serine residues within their aromatic regions, yielding smaller oligomeric species, but the local mechanism by which this occurs had not been defined. Our results show clearly that serine phosphorylation disrupts the interaction in HSPB1, pointing to a shared structural mechanism by which stress-induced phosphorylation disrupts HSPB1 and HSPB5 oligomers, allowing them to form smaller, more dispersed, and more active species (*Lambert et al., 1999*; *Ito et al., 2001*).

Intriguingly, PRE results place residue 83 in the same region where we observe aromatic peptide binding, indicating that the other two phosphorylation sites of HSPB1 (78 and 82) are in proximity to the aromatic region. Several sHSPs have phosphorylation sites in similar sub-regions, and it has been shown for HSPB1 that similar effects on quaternary structure and chaperone activity are obtained from mutation of any of these sites (*Jovcevski et al., 2015*). A recent crystal structure of an HSPB1 construct containing part of the boundary region and the ACD in complex with a peptide containing residues 76–88 phosphorylated at position 82 contained density in this same region. While this density was attributed to the peptide, the identity of the residues could not be resolved (*Collier et al., 2019*). Our results confirm that the two phosphorylation sites could reside near this location. Thus, all three sites of stress-induced modification are near each other and close to a conserved ACD surface that is highly enriched in negatively-charged amino acids (i.e., L3/4 and L5/6), providing an environment that can easily be disrupted by additional negative charge. Both regions were less protected as seen by HDXMS in the phosphorylation-mimicking dimer compared to WT oligomers, consistent with coupled behavior. It has been demonstrated that singly- and doubly-phosphorylated HSPB1 species adopt intermediate-sized oligomers (*Jovcevski et al., 2015*), consistent with the arrangement acting as a rheostat that tunes the distribution of oligomeric states in response to cellular stress. Our results placing sequentially distant phosphorylation sites in spatial proximity demonstrates a common mechanism for the global effects observed for mutation of different phosphorylation sites. While there is clear interplay among the three phosphorylation sites, additional studies will be required to determine the nature of the interactions between the sites in the boundary region and the aromatic region. In particular, the role of boundary region phosphorylation is unclear, as we did not detect a difference in binding by the phosphorylated and non-phosphorylated forms of this peptide. Recent NMR work showed that an elongated ACD construct containing the boundary region with phosphomimetic mutations at regions 78 and 82 formed a transient β2 strand in solution (*Collier et al., 2019*). However, without a direct comparison to a construct of the same length but without phosphomimetic mutations, the role of phosphorylation in this interaction remains unclear.

Our study also provides the first residue-level insights into the effects of known disease-associated mutations within the HSPB1 NTR. The three mutations investigated have previously been shown to alter the oligomeric distribution of HSPB1, in each case yielding larger oligomers (*Muranova et al., 2015*), but an understanding of the effects on a more detailed level are lacking. Remarkably, single mutations in the NTR have profound, widespread effects on dynamics, highlighting sHSP sensitivity to mutation and modification. We find that mutations at residues only five positions apart in the NTR have distinct, almost opposite effects (G34R and P39L) while two mutations that are 50 residues apart from each other (G34R and G84R) produce highly similar effects. In particular, G34R and G84R variants in the conserved and boundary regions respectively each exhibit a coupled increase in deuterium exchange in both the conserved and boundary regions. Furthermore, the mutant G34R conserved region peptide showed a lower affinity for the dimer interface groove. Altogether the results identify an interplay between two non-local regions of the NTR, in which the location of one region affects the other. Both regions can bind at the dimer interface groove, so another way to view the interdependence is that occupancy at a given interface groove by one sub-region favors occupancy by the other.

While the G34R and G84R substitutions are associated with the release of NTR sub-regions from their ACD interactions, P39L-HSPB1 shows markedly increased protection from exchange in the aromatic, conserved, and Trp-rich regions of the NTR. However, there is no large change for the boundary region, suggesting that the interdependence observed for the two glycine-to-arginine mutations is due to substitution of a bulky (and/or charged) amino acid in either the conserved or boundary regions per se. The increased helicity observed in the CD spectrum of P39L-HSPB1 oligomers is consistent with stabilization of helical structure in the Trp-rich region, likely a direct consequence of the helix-favoring proline-to-leucine substitution. Notably, no interactions were detected between the Trp-rich region and the ACD in either the WT- or mutant form, implying that the sub-region could be involved in NTR-NTR interactions. Our inability to make mutations in the Trp-rich sub-region that did not have an impact on oligomer size is further corroboration of the sub-region's central role in driving HSPB1 oligomerization. Indeed, the decreased rate of subunit exchange from P39L-containing oligomers suggests that such NTR-NTR interactions play a rate-limiting role in the dissociation of subunits.

In sum, an approach using solution-state NMR, HDXMS, and modeling has succeeded in defining the heretofore intractable NTR of HSPB1. Rather than behaving as a random coil-like intrinsically disordered region, we find it to be quasi-ordered, with six sub-regions that display distinct properties and binding preferences. The results reveal that, contrary to expectation, the high degree of heterogeneity and polydispersity that is a defining feature of HSPB1 (and other human sHSPs) derives not from fuzzy disorder but rather from an array of combinatorial interactions that involve discrete NTR sub-regions and specific surfaces on the structured ACD. We expect other oligomeric sHSPs are similarly defined and that they can be parsed out using approaches similar to those described here. Finally, it is reasonable to think that there are other examples of quasi-order, with multiple interacting regions in a polypeptide chain, that can likewise be defined at a structural level by similar experimental approaches.

## Materials and methods

### Protein expression and purification

Human HSPB1 (accession # P04792) had previously been cloned into pET23a and pET151d vectors (ampicillin resistant). The B1-ACD construct had previously been optimized to truncation of the full-length sequence from Gln80 to Ser176. Site-directed mutagenesis using the QuikChange protocol was used to introduce substitution mutations throughout the sequence.

Several protocols for protein expression were used to obtain different isotopically labeled samples. In almost all cases (unless otherwise specified) for full-length HSPB1, BL21(DE3) *E. coli* cells were used and a final concentration of 1.0 mM isopropyl β-D-1-thiogalactopyranoside (IPTG) was added to induce protein expression. For the B1-ACD construct, 0.5 mM IPTG was used.

For natural abundance (or non-isotopically labeled) protein, cells were grown in 0.5 L of lysogeny broth (LB) with 100 µg/mL ampicillin. Cells were grown at 37°C in a shaking incubator until $OD_{600}$ ~0.6. IPTG was then added to a final concentration of 1.0 mM and the temperature reduced to 22°C. Protein was expressed in a shaking incubator for ~22 hr. Cells were harvested by centrifugation and resuspension in lysis buffer (50 mM Tris, pH 8.0, 100 mM NaCl, 1 mM ethylenediaminetetraacetic acid [EDTA]).

For $^{15}$N-labeled (no deuteration) protein, MOPS minimal media was used. Per 1 L culture, 1 g of $^{15}$NH$_4$Cl was used for isotopic labeling. 4 g/L of glucose was used. Growth and expression steps were identical to those used for natural abundance protein.

For $^2$H$^{15}$N$^{13}$C-labeled (partial deuteration,~75%) protein, cells were grown in stages to acclimate the cells to deuterated minimal media. For all $^{13}$C-labeling, 3 g/L of $^{13}$C-glucose was used. One colony was grown in 3 mL of LB for ~5 hr and then centrifuged to pellet the cells. These cells were resuspended and grown in 50 mL of H$_2$O-based M9 minimal media to an $OD_{600}$ ~0.6. Cells were again pelleted, resuspended, and grown in 100 mL of D$_2$O-based M9 minimal media to an $OD_{600}$ ~0.6. Cells were then transferred directly to a larger 500 mL (total) D$_2$O-based M9 minimal media and grown to an $OD_{600}$ ~0.6. After induction and reduction of temperature, protein was expressed for ~48 hr.

For $^2$H$^{15}$N$^{13}$C-labeled (perdeuteration) protein, cells were grown in additional stages to acclimate the cells to deuterated minimal media. 3 g/L of $^2$H$^{13}$C-glucose was used for the final culture. Stocks for deuterated M9 minimal media were also prepared in D$_2$O. One colony was grown in 3 mL of LB for ~5 hr and then centrifuged to pellet the cells. These cells were resuspended and grown in 50 mL of H$_2$O-based M9 minimal media to an $OD_{600}$ ~0.6. Cells were again pelleted, resuspended, and grown in 100 mL of D$_2$O-based M9 minimal (non-deuterated glucose) media for 1–2 hr. Cells were again pelleted, resuspended, and grown in 200 mL of D$_2$O-based and $^2$H-glucose-based M9 minimal media to an $OD_{600}$ ~0.6. Cells were then transferred directly to a larger 1 L (total) D$_2$O-based and $^2$H-glucose-based M9 minimal media. Cells were grown to an $OD_{600}$ ~0.6 and induced. After induction and reduction of temperature, protein was expressed for ~48 hr.

Cells containing full-length HSPB1 were lysed by freeze-thaw and incubation with lysozyme and protease inhibitors in lysis buffer on ice for 20 min. Generally 0.5 L cultures of cells were lysed in one tube to maximize yield and purity. Deoxycholate was added to the lysed cells and placed on a shaking incubator at 37°C for 15 min. DNase, RNase, and magnesium chloride were then added and shaking incubation continued for 15 min. Cell lysate was centrifuged at high speed at 4°C.

Ammonium sulfate was added to the supernatant to 40% saturation on a slow shaker at room temperature and allowed to equilibrate for 30 min. Ammonium sulfate precipitate was centrifuged at high speed at 20°C and the supernatant discarded. Pellets were used immediately or stored at −80° C for up to 1 week.

Ammonium sulfate pellets were resuspended in anion exchange buffer (AEX- 20 mM Tris, 10 mM MgCl$_2$, 30 mM NH$_4$Cl, pH 7.6) at room temperature. Remaining solids were pelleted briefly at 20°C. Protein was desalted using a G25 column in AEX buffer at room temperature. Precipitated material was pelleted briefly at 20°C. Desalted protein was separated by an anion exchange DEAE column with a step gradient of AEX buffer with increasing sodium chloride at room temperature. Protein fractions were analyzed for purity by SDS-PAGE, pooled, and concentrated for SEC. Protein was separated on Superdex 200 or 75 columns in 50 mM sodium phosphate (NaPi), 100 mM NaCl, 0.5 mM EDTA, pH 7.5 buffer. Oligomeric proteins (WT, disease mutants) were separated on a Superdex 200 column, while smaller constructs (HSPB1$_{dimer}$, NTR-ACD) were separated on a Superdex 75 column. SEC separated fractions were analyzed for purity by SDS-PAGE, pooled, and concentrated to the desired concentration. Concentration was determined by 280 nm absorbance (extinction coefficient of 40,450 M$^{-1}$cm$^{-1}$).

For cysteine-free proteins or proteins containing only the native cysteine at position 137, no reducing agent was added during purification. With the native cysteine present, the resulting protein was generally >95% oxidized at the dimer interface as seen by dimer formation by non-reducing SDS-PAGE. For proteins containing non-native cysteines (for fluorophore or spin labeling), the reducing agent dithiothreitol (DTT) was included at each stage of purification to avoid disulfide formation.

## Nuclear magnetic resonance spectroscopy

All NMR experiments were carried out on either 600 or 800 MHz Bruker spectrometers equipped with cryoprobes. All samples were prepared in 50 mM sodium phosphate (NaP$_i$), 100 mM NaCl, 0.5 mM EDTA, pH 7.5 buffer. Spectra were collected at 30°C.

Several TROSY-based triple resonance experiments were implemented to assign peaks in the NTR-ACD spectrum to particular residues- HNCO, HN(CA)CO, HNCA, HN(CO)CA, HNCACB, HNCB, and HNCOCANNH ('NNH') experiments. Non-uniform sampling (NUS) at a sampling rate of 25% was used for longer experiments. For NUS datasets, an iterative soft threshold algorithm was used to reconstruct full spectra (*Hyberts et al., 2012*). A maximum protein concentration of 600 µM was used as inter-dimer interactions were evident at higher concentrations (many peaks broadened). Both ~75% deuterated and perdeuterated $^2$H$^{15}$N$^{13}$C-labeled forms of NTR-ACD (expression in previous section) were used for almost all triple resonance experiments, with modest improvements observed for higher deuteration levels. For the HNCB and R2 experiments, only perdeuterated protein was used.

To measure intensities and positions of peaks in 2D spectra, NMRViewJ was used (*Johnson, 2004*). The following equation was used to calculate chemical shift perturbations (CSPs) between peaks in two spectra:

$$CSP = \sqrt{(\delta_H)^2 + (\delta_N/5)^2}$$

Spin-label constructs were made with a NTR-ACD/C137S background and cysteines introduced at positions throughout the NTR, one at a time. Samples were prepared in 50 mM NaPi, 100 mM NaCl, 0.5 mM EDTA, pH 7.5 buffer and 10 mM DTT to fully reduce all cysteines. Reducing agent was then removed from protein samples using a desalting column. Immediately after removing reducing agent, 5-fold molar excess (1-oxyl-2,2,5,5-tetramethylpyrroline-3-methyl)-methanethiosulfonate (MTSL) spin label (in DMSO) was added to protein samples and allowed to incubate overnight at 4°C or at room temperature for 2 hr. Excess MTSL was then removed from the labeled protein using a desalting column. The resulting NMR samples contained 300–400 µM protein. 2D $^1$H-$^{15}$N HSQC-TROSY spectra were collected for each spin-labeled mutant protein. The unpaired spin label was then quenched in each sample by addition of ascorbate (5 mM final concentration). Identical spectra were collected for each quenched sample for intensity comparison between quenched and unquenched spectra.

The distal, aromatic, conserved, and boundary region peptides were purchased from Genscript. The N-terminal residue of all but the distal peptide was formylated, and the C-terminal residue of all

peptides was amidated. The Trp-rich peptides were purchased from LifeTein and were acetylated on the N-terminus and amidated on the C-terminus. The distal, conserved, and boundary peptides were dissolved in NMR buffer prior to use. The aromatic and Trp-rich peptides were dissolved in DMSO to a concentration of 100 or 50 mM, then diluted in buffer to a concentration of 1 mM. All peptides were stored at −80°C. A spectrum of $^{15}$N B1-ACD in the presence of 1% DMSO was collected and used as a reference for the aromatic and Trp-rich peptide experiments, although it was highly similar to the $^{15}$N B1-ACD spectrum in the absence of DMSO.

## Hydrogen-deuterium exchange mass spectrometry

200 μM protein samples were equilibrated at room temperature (22°C) in 50 mM NaPi, 100 mM NaCl, 0.5 mM EDTA at pH 7.5 for several hours. Samples were diluted 10-fold into deuterated buffer (prepared identically but with $D_2O$) for a final concentration of 20 μM and incubated at room temperature for various periods of time to allow for hydrogen-deuterium exchange. At the desired time point, the deuteration reaction was quenched by adding an equal volume of quench buffer (0.6% formic acid) on ice for a final pH of 2.5. Quenched samples were immediately flash frozen in liquid nitrogen and stored at −80°C. Undeuterated samples were prepared in a similar fashion but replaced addition of $D_2O$ buffer with protonated buffer. Fully deuterated samples were made by first denaturing the protein (3M guanidine HCl and high heat for at least 30 min), making the same dilution into $D_2O$ buffer, incubating for several hours, and quenching the same as all other samples.

Samples were stored in liquid nitrogen until 5 min prior to injection to maintain consistent levels of deuterium loss (back-exchange). Initially, the sample was passed over a custom packed pepsin column (1 × 50 mm) at 200 uL/min in 0.1% formic acid at 1°C for digestion of the protein into peptides. Digested peptides were then captured onto a trapping column (Waters vanguard BEH C18 2.1 × 5 mm 1.7 μm 130 Å) and resolved over a C18 reverse-phase column (Waters BEH 1 × 100 mm 1.7 μm 130 Å) using a linear gradient of 3% to 40% B over 10 min (A: 0.1% formic acid, 0.025% trifluoroacetic acid, 2% acetonitrile; B: 0.1% formic acid in acetonitrile). The LC system was coupled to a Waters SYNAPT G2-Si Q-TOF. The source and desolvation temperatures were 70°C and 130°C, respectively. The StepWave ion guide settings were set to minimize non-uniform deuterium loss during desolvation (*Guttman et al., 2016*). The pepsin, trap, and resolving columns were washed extensively to reduce sample carryover (*Fang et al., 2011*; *Majumdar et al., 2012*). The resulting levels of carryover were below 5% for each peptide analyzed based on blank runs. Peptic peptides of WT and mutant proteins were analyzed by tandem MS (MS$^E$) analyzed by ProteinLynx Global SERVER.

Most peptides could be directly compared among all mutants. For peptides containing a mutation the comparisons are only qualitative as the change in exchange could be an effect of both intrinsic exchange and local structure. MassLynx software was used to align spectra of various time points at the appropriate retention times, and HX-Express v2 (*Guttman et al., 2013*) was used to analyze deuterium incorporation and perform bimodal analysis. The deuteration level at each time point was calculated relative to the deuteration levels of the undeuterated and fully deuterated spectra for each peptide. In cases where the fit was very poor due to very broad isotope distributions or clear bimodals, an alternative fitting was used to obtain a bimodal distribution, representing two distinct deuteration states of the peptide at a given time point. For bimodal peptides shown in *Figures 7* and *9*, their behavior was confirmed across several charge states and in most cases several overlapping peptides. Three biological replicates of WT oligomers and HSPB1$_{dimer}$ were examined and showed qualitatively similar patterns in HDX (*Figure 7—figure supplement 2*), with one of these replicates containing the native C137 and including reducing agent to confirm similar behavior to C137S constructs. Disease mutants were examined with single replicates due to limitation of instrument time.

## Modeling

A starting homology model was produced by aligning a crystal structure of the HSPB1 ACD (PDB 4MJH) with a crystal structure of an HSPB2/HSPB3 heterotetramer (PDB 6F2R) and creating a new PDB file with structural elements from both. It contained the HSPB1 ACD, including both β2 strands, a copy of the HSPB2 motif $^{157}$VNEVYISLL$^{164}$ bound in each β4/β8 groove, and a copy of the HSPB2

conserved motif $^{23}$RLGEQRFG$^{30}$ bound in the dimer interface groove. Residues from the HSPB2/HSPB3 crystal structure were mutated to the appropriate sequence for HSPB1 using the PyMOL Molecular Graphics System (Version 2.0, Schrödinger LLC). PyRosetta (*Chaudhury et al., 2010*) was used to eliminate clashes and model in missing segments. The function FastRelax (*Das and Baker, 2008*) was used to relax the starting homology model and eliminate clashes between chains. Fragment insertion using the BlueprintBDR (*Huang et al., 2011*) mover was then used to add the initial residues of the distal region and a segment of the 2/3 loop that was not resolved in the crystal structure. Two distinct connectivities involving the aromatic region were modeled. One structure was generated in which the aromatic region connected the distal region bound in the opposite orientation of the conserved region, such that it wrapped around the edge of the ACD and contacted loops 3/4, 5/6 and 8/9. Another was generated in which it connected the conserved region to the other distal region and crossed over strands β8, β9, and β3. An initial model for each of these two structures was generated using the BlueprintBDR mover, and then subject to 100 rounds of Generalized Kinematic Closure (*Bender et al., 2016*) to generate two sets of 100 structures that sample the conformational flexibility of the aromatic region. Generalized Kinematic Closure was used to model this longer disordered segment because, unlike the BlueprintBDR mover, it does not explicitly bias structures toward a smaller radius of gyration or use stretches of PDB-derived torsion angles. The missing loops connecting the beginning of the NTR to the β2 strands were then modeled in using the BlueprintBDR mover. Again, there were two possible connectivities for each structure: the conserved motif could be connected to either β2 strand, while the distal region not connected to the conserved motif could be connected to the other β2 strand. For each model, both connectivities were sampled, producing four sets with unique topologies that each contained 100 structures. For each step in which additional residues were introduced in the model, FastRelax was used to relax the new segment and the residues adjacent to it. Finally, FastRelax was used to relax all atoms in the final structures.

## Circular dichroism spectroscopy

Samples were prepared in 25 mM NaPi, 50 mM NaCl, and 0.25 mM EDTA buffer at pH 7.5. Samples were incubated at 20 μM at room temperature for several hours prior to measurement. All measurements were made on a Jasco J-1500 CD spectrometer with Peltier temperature control at 20°C, with 1 nm bandwidth, and averaged over three scans. All data were normalized to units of mean residue ellipticity (MRE). Data were collected at 0.1 nm intervals and then smoothed linearly across 1 nm. All data presented have a high-tension voltage below the recommended cutoff for the detector (800 V).

## Fluorescence-based subunit exchange

WT and disease mutant constructs were generated with a cysteine introduced at position 174, analogous to similar fluorescence studies in HSPB5 (*Peschek et al., 2013*), and the native cysteine at position 137 mutated to serine. Protein was buffer exchanged into 50 mM NaPi, 100 mM NaCl, 0.5 mM EDTA, pH 7.5 buffer with 2 mM tris-(2-carboxyethyl)-phosphine (TCEP). Protein was incubated with 3X molar excess of Alexa Fluor 488 maleimide and incubated at 37°C (to facilitate subunit exchange) for at least one hour. Protein was separated from free dye using gravity desalting columns. Fractions collected from desalting columns were analyzed for separation by comparing absorbances at 495 nm (fluorophore extinction coefficient of 73,000 M$^{-1}$cm$^{-1}$) and 280 nm, with an assumed A$_{280}$/A$_{495}$ ratio of 0.11. The resulting pool of fractions was diluted to desired protein concentrations and fluorophore labeling percentages.

Samples were incubated at 37°C at 20 μM and several labeling percentages for at least one hour prior to measurement. For the experiments presented here, samples with 30% fluorophore labeling were mixed 1:2 with unlabeled protein. The resulting dequenching from homo-FRET among fluorophores in oligomers was measured as a function of time. Measurements were collected on a Horiba Fluorolog-3 with double excitation and emission monochromators and Peltier temperature control at 37°C. Excitation and emission wavelengths of 518 nm and 498 nm were used, respectively.

The resulting kinetic data were fitted to the following exponential equation using the R software package to obtain subunit exchange rates:

$$y = A + B * e^{-kt}$$

## Acknowledgements

We thank P Rajagopal and L Tuttle for training and discussions of NMR experiments and J Kang for assistance with HPLC. We thank K Reiter and D Wilburn for critical reading of the manuscript and all members of the Klevit lab for useful discussions and suggestions. This work was supported by NIH grant 2R01 EY017370 to REK, R01GM127579 to MG, NIH T32 GM008268 and Hurd Fellowship in Biophysics from the UW School of Medicine to AFC and HERB, and the Hope Barnes Graduate Fellowship from the UW School of Pharmacy to HERB.

## Additional information

### Funding

| Funder | Grant reference number | Author |
| --- | --- | --- |
| National Eye Institute | R01 EY017370 | Rachel E Klevit |
| National Institute of General Medical Sciences | R01 GM127579 | Miklos Guttman |
| National Institute of General Medical Sciences | NIH T32 GM008268 | Amanda F Clouser Hannah ER Baughman |

The funders had no role in study design, data collection and interpretation, or the decision to submit the work for publication.

### Author contributions

Amanda F Clouser, Conceptualization, Formal analysis, Validation, Investigation, Writing—original draft, Writing—review and editing; Hannah ER Baughman, Conceptualization, Formal analysis, Investigation, Writing—original draft, Writing—review and editing; Benjamin Basanta, Investigation, Methodology, Writing—review and editing; Miklos Guttman, Resources, Formal analysis, Supervision, Methodology, Writing—review and editing; Abhinav Nath, Conceptualization, Supervision, Writing—review and editing; Rachel E Klevit, Conceptualization, Formal analysis, Supervision, Funding acquisition, Investigation, Methodology, Project administration, Writing—review and editing

### Author ORCIDs

Benjamin Basanta https://orcid.org/0000-0003-1118-5269
Rachel E Klevit https://orcid.org/0000-0002-3476-969X

### Decision letter and Author response

Decision letter https://doi.org/10.7554/eLife.50259.025
Author response https://doi.org/10.7554/eLife.50259.026

## Additional files

### Supplementary files

• Transparent reporting form
DOI: https://doi.org/10.7554/eLife.50259.021

### Data availability

NMR resonance assignments have been deposited in BMRB; accession number 27681. Data generated for this study are included in the manuscript and supporting figures and tables. Source data for HDXMS data included in Figures 7 and 9 and associated supplemental tables are provided as Excel spreadsheet.

The following dataset was generated:

| Author(s) | Year | Dataset title | Dataset URL | Database and Identifier |
|---|---|---|---|---|
| Clouser AF, Klevit RE | 2019 | Chemical shift assignments for HSPB1 containing residues 1-176 | http://www.bmrb.wisc.edu/data_library/held.shtml#27681 | Biological Magnetic Resonance Data Bank, 27681 |

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
