## [Decision Letter]

Thank you for submitting your article "Interplay of disordered and ordered regions in a human small heat shock protein yields an ensemble of "quasi-ordered" states" for consideration by *eLife*. Your article has been reviewed by two peer reviewers, including Hannes Neuweiler as the Reviewing Editor and Reviewer #1, and the evaluation has been overseen by Cynthia Wolberger as the Senior Editor. The following individual involved in review of your submission has agreed to reveal their identity: Kristaps Jaudzems (Reviewer #3).

The reviewers have discussed the reviews with one another and the Reviewing Editor has drafted this decision to help you prepare a revised submission.

Review:

The manuscript reports on the investigation of interactions, dynamics and conformation of the intrinsically disordered N-terminal domain (NTR) of the human small heat shock protein (sHSP) HSPB1 using a hybrid approach involving NMR spectroscopy, hydrogen-deuterium exchange mass spectrometry (HDXMS) and molecular modelling. The authors use mutated and truncated protein constructs engineered to form mono-disperse pseudo-wild-type proteins of the otherwise highly heterogeneous, oligomeric wild-type HSPB1. They further report binding studies using a set of peptides covering the sequence area of the NTR to study specific interactions with the folded α-crystallin domain (ACD) of HSPB1. The study involves spin labels and additional, complementary spectroscopy (circular dichroism and fluorescence). The authors conclude that they revealed an ensemble of "quasi-ordered" NTR states that deviate from a common intrinsically disordered protein (IDP), and that they provide new mechanistic insights into the effects of stress and disease related phosphorylation and mutation of the NTR.

IDPs represent a substantial fraction of the human proteome and fulfil critical functions within living cells. However, they are elusive to experimental investigation. Very little is known about their "structure", dynamics and functional mechanisms compared to folded proteins. sHSPs, which are small chaperones that importantly participate in protein homeostasis, consist of a folded and an intrinsically disordered domain. The human sHSP HSPB1 fulfils various chaperoning activities and is involved in severe diseases like cancer. Structural and mechanistic studies of HSPB1 are complicated by intrinsic disorder and by the fact that the active wild-type protein forms a heterogeneous ensemble of oligomers.

The reviewers agree that the study by Clouser et al. provides important new residue-level insights into the disordered NTR and its interactions with the ACD. The authors conclusively show that the NTR, in the context of HSPB1, deviates from a random coil with extensive NTR/ACD and NTR/NTR contacts as well as some residual structure. This is evident from analysis of NMR chemical shifts, line broadening and the observation of multiple peaks for one and the same residue reflecting conformationally distinct, populated states. HDXMS experiments reveal changes in solvent-accessibility in the NTR and their response to phosphorylation and mutation. The comprehensive set of experiments is well designed, thoroughly conducted and analysed. The molecular models inferred from experimental results help to structurally visualize the results.

The manuscript is logically structured, well written and results are clearly presented and discussed. However, a major concern surrounds the nature and definition of the "quasi-ordered" states the authors claimed to have identified (highlighted in title and Abstract).

Essential revisions:

1) The authors conclude that the nature of "quasi-ordered" states they revealed for the NTR would deviate from that of a common IDP and would thus represent new and previously unknown IDP characteristics. This conclusion appears to be an over-interpretation of data. If one would study the NTR in isolation the domain would probably show all the characteristics of a common IDP (can the NTR be expressed and studied in isolation?). In the second paragraph of the Discussion the authors state that given the NTR is an IDP, the specific interactions detected with the ACD were surprising. They further imply (Discussion, third paragraph) that according to their results the NTR cannot be classified as IDP. But the system investigated by the authors is not common in the sense that the NTR is covalently linked to its binding partner (ACD). Since one of the functions of the NTR is to oligomerize HSPB1 the NTR evolved to form inter-molecular interactions with ACDs of other HSPB1 proteins. Interactions of the NTR with the ACD are thus expected. Inter-molecular NTR-ACD interactions are likely the same or similar as intra-molecular NTR-ACD interactions revealed by the authors. The latter ones are of mono-molecular nature and thus entropically favoured compared to the inter-molecular ones. Fusing an IDP to its binding partner using recombinant methods may emerge as a new and interesting approach to study weak interactions of IDPs with binding partners.

2) A weakness of the study is that the dynamics of NTR sub-regions (assigned parts) and their effects on ACD dynamics is not evaluated. This makes it difficult to assess whether the identified states are really quasi-ordered, fuzzy or transient. In the second paragraph of the subsection “Conformational heterogeneity is observed throughout the NTR”, the authors state that the boundary sub-region has random-coil chemical shifts, which does not seem to fit with it being "quasi-ordered". Furthermore, the authors write "For this reason, we propose the term "quasi-ordered" to describe the NTR of HSPB1, as it makes highly-specific long-lived contacts while remaining dynamic and heterogeneous", but don't provide sufficient evidence for "long-lived contacts while remaining dynamic". Therefore, the authors should additionally perform an NMR dynamics study or at least summarize (from their existing data) the exchange of each sub-region (both on NTR as well as ACD) with respect to NMR timescale.

3) On a similar note: the authors state: "…two conformational states of random coil segments move slowly…": can the authors provide a rough estimate or lower limit of the interconversion time constant? This would be interesting information because backbone dynamics of random coil peptides are known to be ultrafast (nanosecond time scale).

4) In their peptide binding experiments the authors apply 1 mM peptide concentrations, which is excess over protein, and observe effects below saturation (Figure 4A, subsection “Distinct regions of the NTR bind different ACD interfaces”, fourth paragraph). This indicates non-specific, non-cooperative binding. If high mM concentrations are required for binding one can hardly argue for specific interactions. Binding equilibria with dissociation constants >1 mM are commonly classified as non-specific. In contrast, the addition of NTR-ACD to B1-ACD (mixed dimer) yields overlapping spectra (Figure 2A).

5) Figure 4A: The changes of chemical shifts towards the NTR-ACD values, observed upon peptide binding, are not evident for all peaks indicated by arrows: Signals for residue 113 are unclear; signals for residue 155 move in another direction than indicated.

6) In the Abstract and Discussion section the authors claim that their results regarding phosphorylation-dependent interactions would inform a mechanism. But what precisely is the new mechanistic information their data provide over the already existing knowledge (Lambert et al., 1999; Ito et al., 2001)?

7) Discussion section: The effects induced by the remote mutations G34R and G84R are likely to be similar because in both cases a hydrogen is replaced by a bulky, positively charged side chain, which will induce steric and electrostatic repulsion (observed by release of NTR sub-regions). G34R and P39L show very different effects because the nature of these mutations is very different. Proline is known to be a helix breaker, which is removed by mutation P39L. This explains enhanced helix content and interaction.

---

## [Author Response]

Essential revisions:1) The authors conclude that the nature of "quasi-ordered" states they revealed for the NTR would deviate from that of a common IDP and would thus represent new and previously unknown IDP characteristics. This conclusion appears to be an over-interpretation of data. If one would study the NTR in isolation the domain would probably show all the characteristics of a common IDP (can the NTR be expressed and studied in isolation?). In the second paragraph of the Discussion the authors state that given the NTR is an IDP, the specific interactions detected with the ACD were surprising. They further imply (Discussion, third paragraph) that according to their results the NTR cannot be classified as IDP. But the system investigated by the authors is not common in the sense that the NTR is covalently linked to its binding partner (ACD). Since one of the functions of the NTR is to oligomerize HSPB1 the NTR evolved to form inter-molecular interactions with ACDs of other HSPB1 proteins. Interactions of the NTR with the ACD are thus expected. Inter-molecular NTR-ACD interactions are likely the same or similar as intra-molecular NTR-ACD interactions revealed by the authors. The latter ones are of mono-molecular nature and thus entropically favoured compared to the inter-molecular ones. Fusing an IDP to its binding partner using recombinant methods may emerge as a new and interesting approach to study weak interactions of IDPs with binding partners.

We apologize for any lack of clarity – we agree that the NTR should still be classified as an intrinsically disordered region (IDR) and have updated the text in the Discussion to clarify this point. By introducing the term “quasi-order”, we are not attempting to de-classify the NTR of HSPB1 as an IDR but rather aim to further the conversation about IDPs/IDRs and expand the classification of their behaviors. Terms such as “fuzzy complex” and “folding-upon-binding” have been previously used to describe interactions between disordered proteins and their binding partners. However, the NTR-ACD interactions described here do not fit well into either category. They are orientationally specific (i.e. the segments of the NTR that interact with the ACD tend to adopt single structural states with a defined orientation in relation to their binding grooves) and thus cannot be described as fuzzy. Nevertheless, NTR-ACD interactions remain dynamic and do not form static structures and thus cannot be said to fold upon binding. We propose the term quasi-ordered to describe the interactions as they do not fit into the existing framework described in the IDP literature. It seems likely that other IDPs or IDRs may also fit into this category and that it is therefore useful to the field to define a term to describe this phenomenon.

We agree that the fact that the NTR interacts with the ACD is not itself surprising. The novelty (and our associated surprise) is in discovering a tractable, high resolution approach to characterize the IDR in a system where historically the IDR (NTR of sHSPs) has been truncated or described only in very coarse-grained terms. The NTR is enriched in hydrophobic amino acids relative to other disordered proteins and is highly susceptible to proteolytic degradation, rendering it difficult to generate and study in isolation. Furthermore, given the extensive interactions between the NTR and ACD our studies have uncovered, we question whether the behavior of the NTR in isolation would be physiologically relevant. Based on the properties of its sequence, we predict it to be collapsed or molten globule-like, but we have not tested this experimentally because we are doubtful that the experimental results outside its native molecular context will shed light on its biological function.

The extensive, specific, but variable-in-combination interactions between the NTR and ACD provide an essential view of how sHSPs regulate availability of their NTRs for oligomerization or client interactions. We have changed phrasing in the Discussion to reflect that the fact of NTR-ACD interactions itself is not surprising, but rather that the nature of these interactions is surprising and crucial to understanding how sHSPs function.

Changes to address these points are included in the first two paragraphs of the Discussion.

2) A weakness of the study is that the dynamics of NTR sub-regions (assigned parts) and their effects on ACD dynamics is not evaluated. This makes it difficult to assess whether the identified states are really quasi-ordered, fuzzy or transient. In the second paragraph of the subsection “Conformational heterogeneity is observed throughout the NTR”, the authors state that the boundary sub-region has random-coil chemical shifts, which does not seem to fit with it being "quasi-ordered". Furthermore, the authors write "For this reason, we propose the term "quasi-ordered" to describe the NTR of HSPB1, as it makes highly-specific long-lived contacts while remaining dynamic and heterogeneous", but don't provide sufficient evidence for "long-lived contacts while remaining dynamic". Therefore, the authors should additionally perform an NMR dynamics study or at least summarize (from their existing data) the exchange of each sub-region (both on NTR as well as ACD) with respect to NMR timescale.

Our existing data support the idea that many of the NTR-ACD interactions we describe are long-lived. For most observable exchange behavior, the timescale is very slow (several hundred milliseconds to several minutes). In the NMR studies, multiple peaks were observed for several residues in both the NTR and ACD. The doubled ACD peaks arise from residues that show contacts with the NTR. This “slow exchange” behavior places an upper limit on the exchange rate between states. In response to the reviewers’ request for more information regarding the timescale of the dynamics, we calculated the frequency differences for all pairs of NMR peaks that show doubling. The smallest difference is ~2 Hz, implying a lifetime greater than ~500 msec. We have clarified this point in the fourth paragraph of the subsection “The disordered NTR makes extensive contacts with the ACD” and in the second paragraph of the subsection “Conformational heterogeneity is observed throughout the NTR”.

Additionally, we collected ^15^N R2 data on the NTR-ACD construct. The data reveal random-coil like dynamics for the NTR insertion sub-region and faster relaxation for Trp-rich and boundary NTR sub-regions, comparable to slower motions of the structured ACD. These data, presented as a new panel in Figure 6 (6D) and discussed in the fourth paragraph of the subsection “Conformational heterogeneity is observed throughout the NTR”,are congruent with the conclusions from the HDXMS results.

Although as mentioned (subsection “Conformational heterogeneity is observed throughout the NTR”, eighth paragraph), our HDXMS data do not allow us to extract precise rates of exchange, the bimodal behavior observed is still informative for estimating lifetimes. The distal and boundary regions show distinct populations at 1 minute and 4 minutes, respectively, indicating lifetimes on the order of minutes for states involving contact of these regions with the ACD in the oligomeric context.We added text clarifying this point to the sixth and seventh paragraphs of the subsection “Conformational heterogeneity is observed throughout the NTR” and in the third paragraph of the Discussion.

3) On a similar note: the authors state: "…two conformational states of random coil segments move slowly…": can the authors provide a rough estimate or lower limit of the interconversion time constant? This would be interesting information because backbone dynamics of random coil peptides are known to be ultrafast (nanosecond time scale).

Following from our response to comment 2, we can estimate the exchange between states of the insert region to be in the regime of tens of milliseconds or slower (smallest observed chemical shift difference is 24 Hz). The two states cannot be distinguished in the HDX data, as they are both highly solvent-exposed and have exchanged fully at the first timepoint. The changes we made in response to point 2 should address this comment as well.

4) In their peptide binding experiments the authors apply 1 mM peptide concentrations, which is excess over protein, and observe effects below saturation (Figure 4A, subsection “Distinct regions of the NTR bind different ACD interfaces”, fourth paragraph). This indicates non-specific, non-cooperative binding. If high mM concentrations are required for binding one can hardly argue for specific interactions. Binding equilibria with dissociation constants >1 mM are commonly classified as non-specific. In contrast, the addition of NTR-ACD to B1-ACD (mixed dimer) yields overlapping spectra (Figure 2A).

We respectfully disagree that the interactions observed in the peptide binding studies are non-specific. A distinguishing feature of IDPs is their ability to engage in specific yet low-affinity interactions. Each peptide perturbs specific localized regions of the ACD. These perturbed regions are in good agreement with the sites of contact observed in the PRE experiments in the context of the NTR-ACD construct. If the peptide binding results were non-specific, we would expect to see effects in regions of the ACD that are not in agreement with the PRE results, which does not occur. This fits our notion of a specific interaction.

Secondly, while the peptide-ACD interactions are low affinity in *trans*, the effective local concentration of the NTR when it is physically attached to the ACD will be fairly high, such that concentrations approaching 1mM are relevant. This contention is borne out in the mixed dimer experiment, where the effects of NTR/ACD contacts from a tethered NTR (in *trans*) recapitulate the chemical shifts observed in the full-length dimer. Similarly, the concordance of the PRE results involving the full, tethered NTR with those of the peptide binding experiments create a uniform set of observations.

We have added a comment regarding the effective local concentration of an attached NTR in the fourth paragraph of the subsection “Distinct regions of the NTR bind different ACD interfaces”.

5) Figure 4A: The changes of chemical shifts towards the NTR-ACD values, observed upon peptide binding, are not evident for all peaks indicated by arrows: Signals for residue 113 are unclear; signals for residue 155 move in another direction than indicated.

We have revised this figure for clarity. The peaks corresponding to the ACD and

NTR-ACD constructs are now labeled in black and grey, respectively. For residue 113, we labeled the two intermediate (peptide titration) peaks as well. Because this residue has the highest CSP, these peaks are relatively low in intensity and may have been easy to miss. The arrow for residue 155 was removed.

6) In the Abstract and Discussion section the authors claim that their results regarding phosphorylation-dependent interactions would inform a mechanism. But what precisely is the new mechanistic information their data provide over the already existing knowledge (Lambert et al., 1999; Ito et al., 2001)?

Previous work (cited in our manuscript) has established that phosphomimetic mutations at each of the three sites in HSPB1 have similar global effects on oligomeric size and chaperone activity, despite the fact that these sites are not all close in sequence. Which site is modified seems to matter less than the total number of phosphorylated sites for the aforementioned effects. Our work provides a structural explanation for this phenomenon, by defining the first phosphorylation-dependent NTR/ACD interaction and showing that the phosphorylation site in the aromatic region (Ser 15) is close in space to the sequentially-distant sites Ser78 and Ser82. These sites all interact with the same region of the ACD, specifically the dimer interface groove and β5/6 loop, As we point out in the Discussion, the known phosphorylation sites in other human sHSPs reside within the same NTR sub-regions, suggesting a common mechanism.

We have added clarifying points in the sixth and seventh paragraphs of the Discussion.

7) Discussion section: The effects induced by the remote mutations G34R and G84R are likely to be similar because in both cases a hydrogen is replaced by a bulky, positively charged side chain, which will induce steric and electrostatic repulsion (observed by release of NTR sub-regions). G34R and P39L show very different effects because the nature of these mutations is very different. Proline is known to be a helix breaker, which is removed by mutation P39L. This explains enhanced helix content and interaction.

We are unsure of the action point the reviewers are requesting. We state the behavior of each type of mutation in both the Results and Discussion sections. While the G34R and G84R mutations may be expected to have similar effects within their immediate proximity, given their sequential distance it was not obvious that they would have similar global effects. The novel finding is that the same structural effects are observed for both mutations, which is explained by their similar contexts of placement in the dimer interface groove. We state explicitly in the Discussion that “The increased helicity observed in the CD spectrum of P39L-HSPB1 oligomers is consistent with stabilization of helical structure in the Trp-rich region, likely a direct consequence of the helix-favoring proline-to-leucine substitution,” which is in agreement with your final point.